# Cell-type-specific plasticity of inhibitory interneurons in the rehabilitation of auditory cortex after peripheral damage

Manoj Kumar [1], Gregory Handy [2], Stylianos Kouvaros [1], Yanjun Zhao[1], Lovisa Ljungqvist Brinson[1], Eric Wei[1], Brandon Bizup[1], Brent Doiron[2] & Thanos Tzounopoulos [1]

Peripheral sensory organ damage leads to compensatory cortical plasticity that is associated with a remarkable recovery of cortical responses to sound. The precise mechanisms that explain how this plasticity is implemented and distributed over a diverse collection of excitatory and inhibitory cortical neurons remain unknown. After noise trauma and persistent peripheral deficits, we found recovered sound-evoked activity in mouse A1 excitatory principal neurons (PNs), parvalbumin- and vasoactive intestinal peptide-expressing neurons (PVs and VIPs), but reduced activity in somatostatin-expressing neurons (SOMs). This cell-type-specific recovery was also associated with cell-type-specific intrinsic plasticity. These findings, along with our computational modelling results, are consistent with the notion that PV plasticity contributes to PN stability, SOM plasticity allows for increased PN and PV activity, and VIP plasticity enables PN and PV recovery by inhibiting SOMs.

In all sensory systems, damage to peripheral organs leads to compensatory cortical reorganization and increased cortical sensitivity to the non-damaged (spared) sensory input[1–9]. This plasticity is crucial for survival, for it is associated with a remarkable recovery of perceptual capabilities[10,11]. Despite the great impact of this plasticity, the underlying system, circuit, and cellular mechanisms remain poorly understood. The establishment of these mechanisms will reveal major concepts in cellular and functional cortical recovery after peripheral damage. Moreover, it holds the promise to highlight novel strategies for enhancing perceptual recovery and mitigating brain disorders associated with sensory deficits and subsequent maladaptive cortical plasticities, such as schizophrenia, tinnitus, phantom limb pain, and neuropathic pain[12–16].

In the auditory system, while the auditory nerve input to the brainstem is significantly reduced after cochlear damage caused by loud noise or ototoxic compounds, cortical sound-evoked activity is maintained or even enhanced[10,11,16–21]. This recovery is associated with increased cortical gain, the slope of cortical responses against sound levels[10,11,21–23]. This plasticity (recovery) is also associated with the

recovery of perceptual sound-detection thresholds after cochlear damage[10,11,20,22,23]. While this association will not be studied here, we will study the underlying cortical mechanisms of this plasticity.

In terms of underlying mechanisms, it is known that the increased cortical gain is associated with reduced inhibitory (GABAergic) cortical activity, increased spontaneous firing, and reorganization of frequency tuning toward less damaged regions of the cochlea[6,10,18,21,24–27]. Moreover, a steep drop in PV-mediated inhibition to principal neurons (PNs) is a predictor of auditory cortical response rehabilitation after cochlear nerve damage[21]. Although the role of a general or PV-centric reduced inhibition is well documented[11,21,25,27–31], it does not provide the precise cellular and circuit mechanisms that mediate cortical rehabilitation. Moreover, the recovery of cortical sound processing after noise trauma is likely also influenced by subcortical plasticity mechanisms[10,32], which will not be studied here. Here, we will study the plasticity of different cortical neuronal subtypes after noise trauma.

The recent use of cell-type-specific labeling and optogenetic manipulations, combined with the genetic and physiological dissection of cortical interneurons (INs)[33,34], have established a new picture of

[1]Pittsburgh Hearing Research Center, Department of Otolaryngology, University of Pittsburgh, Pittsburgh, PA 15261, USA. [2]Departments of Neurobiology and Statistics, University of Chicago, Chicago, IL 60637, USA. ✉e-mail: Mak328@pitt.edu; Thanos@pitt.edu

our understanding of cortical circuits. The canonical cortical circuit includes (at a minimum) VIP, SOM, and PV INs, all with distinct and sequentially organized synaptic connections among themselves and PNs[35–37]. This circuit design begs for a more precise mechanistic understanding of how specific cortical gain modulations associated with an overall and non-specific decrease in inhibition, such as increased cortical gain after peripheral trauma, are implemented and distributed over these distinct IN subclasses. Namely, cortical inhibition is crucial for suppressing neuronal activity[38–41], firing rate gain modulation[38,42–46], and spike timing control[47,48], as well as for correlated neuronal[49,50] and population activity[51,52]. Cortical inhibition is also essential for the prevention of runaway cortical activity that would otherwise lead to pathologic activity[40,53,54]. As such, this complex role of inhibition is expected to pose constraints on how reduced inhibition can safely modulate cortical gain, for a global and non-specific inhibitory reduction could lead to instability and pathology, such as epileptic-like activity[54].

To study the precise mechanisms of inhibition in cortical plasticity after peripheral damage, we used a mouse model of noise-induced cochlear damage. We employed electrophysiological and immunohistochemical assays to assess peripheral damage, longitudinal in vivo two-photon (2P) calcium imaging to assess the activity of different cortical neuronal subtypes, ex vivo electrophysiology assays to assess cellular excitability of different cortical neuronal subtypes, and computational models to shape our hypotheses and predictions. Our results demonstrate that the recovery of cortical sensory processing after peripheral damage is supported by remarkable cell-type-specific plasticity among multiple IN cortical subtypes.

## Results

### Reduced sound-evoked activity of auditory nerve after NIHL

To cause peripheral damage, we used a noise-induced hearing loss (NIHL) paradigm. Mice were bilaterally exposed to an octave band (8–16 kHz) noise at 100 dB SPL for 2 h (Fig. 1a, b). To assess the consequences of this noise exposure on peripheral structures, we measured and quantified the auditory brainstem response (ABR) before and 1, 3, and 10 days after noise exposure. ABR represents the sound-evoked action potentials generated by the synchronized activity of various nuclei of the auditory pathway from the auditory nerve to the brainstem, where ABR wave 1 represents the sound-evoked synchronized activity of the auditory nerve (AN) type-I spiral ganglion neurons (SGNs) (Fig. 1c). We found that noise exposure increased the ABR threshold, the sound-level which elicited a significant wave 1 amplitude (Fig. 1c–e), and reduced the gain of the AN sound-evoked activity, the slope of ABR wave I amplitude against sound level, (Fig. 1f, g). Further, we found that noise exposure significantly increased the ABR response thresholds for 8–32 kHz tones, which remained elevated even 10 days after noise exposure (Fig. 1h–j), suggesting persistent and widespread cochlear damage across the tested frequencies. We also observed frequency-specific damage 10 days after noise exposure, where we found a larger ABR threshold shift at 32 kHz compared to 8 kHz tones (Fig. 1, Supplementary 1a). Moreover, we found that noise exposure increased the distortion product otoacoustic emissions (DPOAE) threshold (Fig. 1k), suggesting a dysfunction of the cochlear outer hair cells (OHCs) sound amplification role. ABR threshold and gain represent the combined functionality of inner hair cells (IHCs), OHCs, type-I SGNs, and synapses between the IHCs and type-I SGN dendrites called ribbon synapses[55]. To identify potential anatomical contributions to the reduced AN gain and elevated ABR and DPOAE thresholds, we performed immunohistochemical analysis across the tonotopic axis of the cochlea to quantify the survival of IHCs, OHCs, and the number of ribbon synapses (Fig. 1l, m and Supplementary Fig. 1b). We found that noise exposure significantly reduced the number of ribbon synapses per IHC in the high-frequency region (16–32 kHz) of the cochlea (Fig. 1l, m), without affecting the survival of either IHCs or OHCs

(supplementary Fig. 1b–d). We did not observe any changes in sham-exposed mice, which underwent identical procedures but without the presentation of sound (Fig. 1 and Supplementary Fig. 1). Together, our noise trauma protocol, by reducing the AN input and increasing peripheral hearing thresholds, reduces the amount and transfer of peripheral auditory input to the brain. We will use this protocol and the same time points after noise trauma to assess the cellular mechanisms of cortical recovery after peripheral damage.

### Robust PN sound-evoked activity (recovery) after NIHL

Next, to study the cellular and circuit mechanisms underlying cortical plasticity after noise trauma, we investigated the sound-evoked responses and the intrinsic excitability properties of the different neuronal subtypes residing in the primary auditory cortex (A1) at different time points for 10 days after noise- or sham-exposure. We first investigated the sound-evoked activity of A1 PNs at 1, 3, and 10 days after exposure (Fig. 2). To selectively image sound-evoked responses from populations of PNs, we used adeno-associated virus (AAV) driven by the calcium/calmodulin-dependent protein kinase 2 (CaMKII) promoter to express the genetically encoded calcium indicator GCaMP6f (AAV-CaMKII-GCaMP6f) in putative PNs (Fig. 2a, b). Twelve to 16 days after stereotaxic viral injections of GCaMP6f (Fig. 2a), we employed acute in vivo wide-field transcranial fluorescent imaging in head-fixed unanesthetized (awake) mice (Fig. 2b). After localizing A1 (Methods), we presented broadband sounds (6–64 kHz, 100 ms long) at 30–80 dB SPL and imaged the sound-evoked changes in the A1 GCaMP6f fluorescence ($\Delta F/F\%$) (Fig. 2c). Each sound was presented 8–10 times in a pseudo-random order. We first measured PNs' response threshold, the sound level which elicits a significant response. Consistent with the increased ABR wave I response threshold, identified as the AN threshold, we found that the PNs' response threshold was significantly increased 1 and 3 days after NIHL (Fig. 2d). However, 10 days after NIHL, PNs' response threshold was significantly lower than the AN threshold (Fig. 2d right). Next, we measured the amplitudes of sound-evoked responses of A1 PNs (Fig. 2e). We found that PN response amplitudes were reduced 1 day after NIHL (Fig. 2e, red), but showed significant recovery in 3 and 10 days after NIHL (Fig. 2e, cyan), and even surpassed pre-noise-exposed response amplitudes in response to suprathreshold sound levels (at 75 and 80 dB SPL). We next quantified the response gain of sound-evoked activity of A1 PNs (Fig. 2f). We calculated gain as the average change in the fluorescence signals ($\Delta F/F\%$) per 5 dB SPL step starting from response threshold[10,11,20]. In contrast to ABR wave I response gain (AN gain), which remained decreased after noise trauma (Fig. 2f, light gray), PN gain was increased and remained increased during the 10 days after NIHL (Fig. 2f, dark gray), which is consistent with previous results[10,16,17]. Moreover, we did not find any changes in response threshold, response amplitude, and gain in sham-exposed mice (Supplementary Fig. 2b–d). Together, these results suggest that 10 days after peripheral damage, A1 PNs display increased gain and recovered response thresholds and amplitudes, despite the persistent peripheral damage.

Wide-field imaging reflects neuronal responses arising from different neuronal compartments (e.g., somata, dendrites, and axons) and different cortical layers[56], thus providing a helpful overall assessment of A1 evoked responses after noise trauma. Moreover, wide-field imaging reflects responses from a population of neurons, but individual neurons may have distinct sound-evoked responses (e.g., recovered vs. non-recovered) after NIHL. Thus, to obtain a more detailed account of individual neuronal responses, we performed longitudinal 2P imaging for 10 days after NIHL (Fig. 2g–r, and Supplementary Fig. 2e–j). After locating A1, we presented trains of broadband sounds and imaged the sound-evoked responses of individual A1 L2/3 PNs' somata (Fig. 2i, j). To use each neuron as its own control, we tracked the same individual A1 L2/3 PNs for 10 days after NIHL (Fig. 2i; "Methods"). Pre-exposure sessions lasted two days, and average

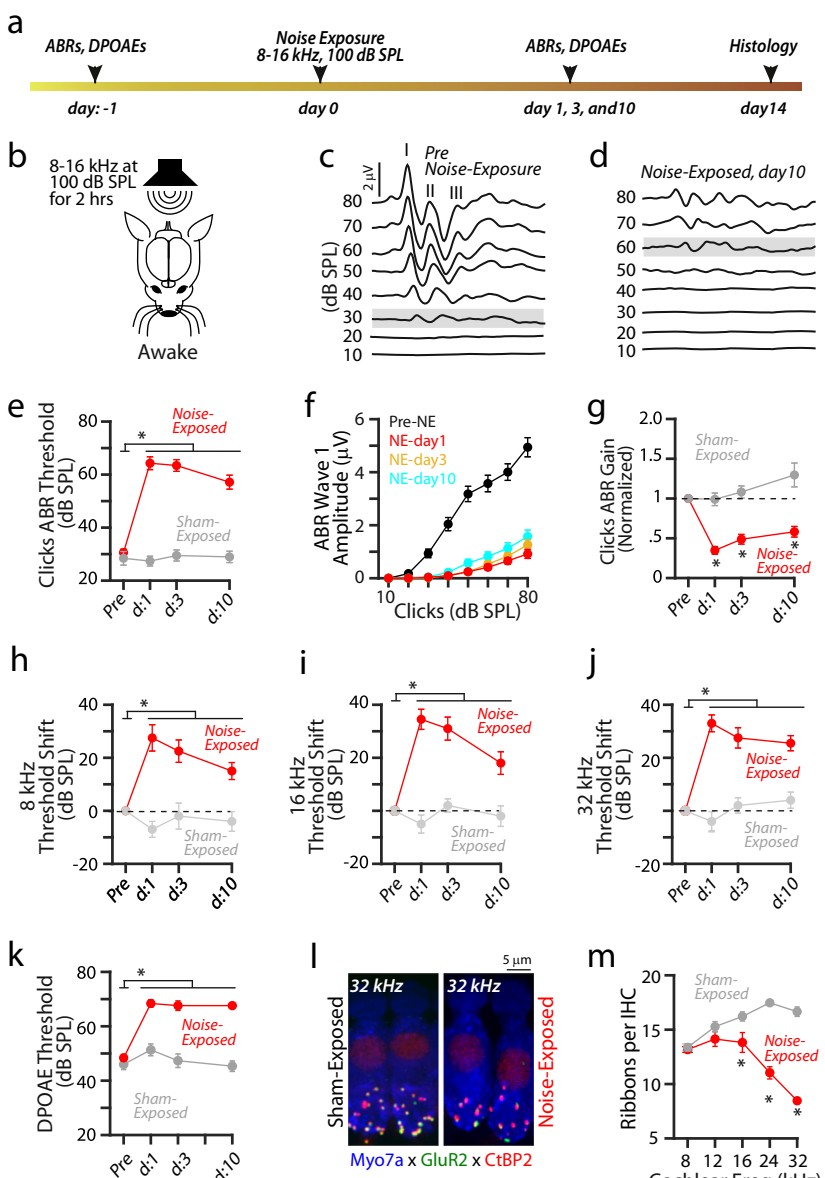

**Fig. 1 | Reduced sound-evoked activity of auditory nerve after NIHL. a** Timetable of experimental design. **b** Noise-exposure paradigm. **c, d** Representative ABR traces clicks before and 10 days after noise exposure. **e** Average ABR thresholds from noise-exposed ($n = 35$) and sham-exposed ($n = 19$) mice before- and at 1, 3, and 10 days after exposure. (Noise vs. sham: effect of exposure, $F = 221.3$, $p = 1.4 \times 10^{-34}$). **f** Average ABR wave 1 amplitude to clicks before and after noise exposure (effect of noise-exposure, $F = 227.7$, $p = 4.9 \times 10^{-38}$). **g** Average ABR gain from noise- and sham-exposed mice, before- and at 1, 3, and 10 days after exposure. (Noise vs. sham: effect of exposure, $F = 56.4$, $p = 7.0 \times 10^{-10}$). **h** Average ABR thresholds shift to 8 kHz tone from noise-exposed ($n = 20$) and sham-exposed ($n = 10$) mice, before- at 1, 3, and 10 days after exposure (Noise vs. sham: effect of exposure, $F = 11.6$, $p = 0.001$).

**i** Same as **h** but for 16 kHz (Noise vs. sham: effect of exposure, $F = 21.8$, $p = 6.7 \times 10^{-5}$). **j** Same as **h** but for 32 kHz (Noise vs. sham: effect of exposure, $F = 8.8$, $p = 6.0 \times 10^{-3}$). **k** Average DPOAE thresholds from noise-exposed ($n = 5$) and sham-exposed mice ($n = 4$) before- and at 1, 3, and 10 days after exposure (Noise vs. sham: effect of exposure, $F = 67.9$, $p = 1.0 \times 10^{-10}$). **l** Cochlear histology images of a 32 kHz frequency region from noise-exposed mice showing reduced ribbon synapses onto inner hair cells (blue) compared to sham-exposed mice. The CtBP2 (red) and GluR2 (green) are pre- and postsynaptic markers, respectively. **m** Average ribbon synapses onto per inner hair across the tonotopic region of the cochlea from noise-exposed and sham-exposed mice (Noise vs. sham, effect of exposure, $F = 126.3$, $p < 1.0 \times 10^{-10}$). $*p < 0.05$; statistical values are given in Supplementary Table 1.

responses of individual neurons from both days were used as pre-exposure responses. After a motion and neuropil correction ("Methods"), we were able to track 531 L2/3 PNs from 11 mice for 10 days after NIHL. To identify the sound-responsive neurons, we used a tone sensitivity index ($d'$), which reflects the neurons' selectivity for preferred frequency against non-preferred frequency[57,58] ("Methods"), and only the neurons with $d' \geq 0$ were analyzed further ($n = 358/531$ PNs from 11 mice). Consistent with our wide-field imaging results, we found that the response thresholds of individual L2/3 PNs were fully recovered 10 days after NIHL and had a similar cumulative distribution of

response thresholds compared to pre-noise-exposure thresholds (Fig. 2k, l). Also, we found that the sound-evoked responses of individual PNs were reduced 1 day after NIHL but overall recovered or even surpassed pre-noise-exposure responses 10 days after NIHL (Fig. 2m). Also, consistent with our wide-field imaging results, the gain of individual PNs was increased after NIHL and remained elevated even 10 days after NIHL (Fig. 2n), showing a shift in cumulative distribution towards higher gain (Fig. 2o). When we plotted individual PN gain after noise-exposure against pre-noise-exposure gain, we also found that on average the gain was increased after noise exposure (Fig. 2p) and the

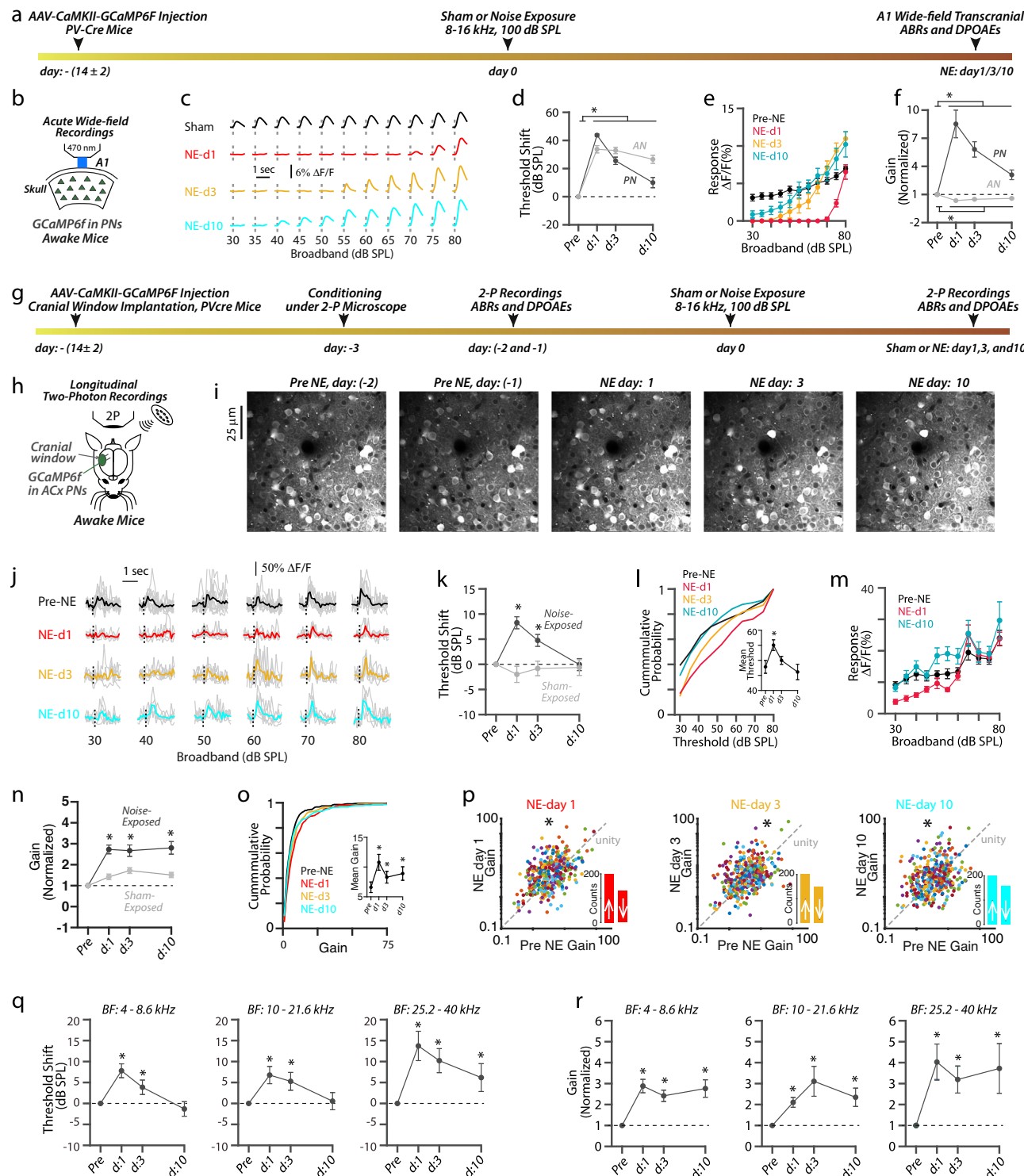

**Fig. 2 | Robust PN sound-evoked activity (recovery) after NIHL. a** Wide-field (WF) imaging experimental design for A1 PNs. **b** Experimental setup. **c** Representative transcranial fluorescence responses of A1 PNs from sham- and noise-exposed mice. **d** Average change in A1 PN response thresholds (dark gray) before and at 1, 3, and 10 days after noise exposure. The average change in AN threshold (light gray) reproduced from Fig. 1. **e** Average sound-evoked responses of A1 PNs to broadband sounds from noise-exposed mice. (Intensity vs. time, the effect of time, $F = 66.18$, $p = <1.1 \times 10^{-10}$). **f** Average response gain of PNs (dark gray) normalized to pre-noise-exposed gain and at 1, 3, and 10 days after noise exposure (effect of exposure, $F = 63.5$, $p = 3.8 \times 10^{-6}$). Normalized AN gain (light gray) reproduced from Fig. 1. **g** Timetable of longitudinal 2P imaging experimental design for A1 L2/3 PNs. **h** Experimental setup illustrating 2P imaging. **i** Z-stack images of tracked PNs. **j** Representative responses from a PN before and after NIHL. **k** Average change in

response threshold of individual PNs from noise (dark gray) and sham (light gray) exposed mice (Noise-exposed: 358 PNs from 11 mice, sham-exposed: 218 PNs from 5 mice, noise vs. sham: effect of exposure, $F = 11.6$, $p = 6.9 \times 10^{-4}$). **l** Cumulative probability of PN response threshold. Inset: Average mean threshold of PNs per mouse. **m** Average responses of individual PNs. **n** Average gain of individual PNs normalized to pre-exposed gain from noise (dark gray) and sham (light gray) mice. **o** Cumulative probability of PN gain. Inset: Average mean gain of PNs per mouse. **p** Scatter plots of the gain of individual A1 L2/3 PNs before and after NIHL. The dotted line represents unity. Insets: Bar graphs represent the number of neurons showing increased gain (↑ above unity) and reduced gain (↓ below unity) after NIHL. **q** Average change in response threshold and gain **r** of PNs with Low-BF (left, $n = 175$ PNs), Mid-BF (middle, $n = 110$ PNs), and High-BF (right, $n = 47$ PNs). *$p < 0.05$; statistical values are given in Supplementary Table 2.

majority of PNs showed increased gain after NIHL (Fig. 2p insets and Supplementary Fig. 2j: day 1: 228/358, day 3: 208/358, and day 10: 199/358).

To correlate the cochlear damage with the plasticity in the sound-evoked activities of individual L2/3 neurons, we analyzed the response threshold and gain of PNs as a function of their pre-exposure best frequencies (BF), the sound frequency with the maximal sound-evoked response[59] (Fig. 2q, r; "Methods"). Irrespective of their BF, PNs showed increased response thresholds 1 day after NIHL (Fig. 2q). However, 10 days after NIHL, PNs with low- (4-8.6 kHz) and mid-BF (10-21.6 kHz) showed complete recovery in their response thresholds, but PNs with high-BF (25.2–40 kHz) did not (Fig. 2q). Consistent with previous results[20], our results support the notion that PNs with BFs corresponding to the high-frequency cochlear region, which showed more damage compared to the low-frequency region (Fig. 1, Supplementary 1a), do not recover their response thresholds completely. In terms of gain, irrespective of their BF, PNs showed increased gain 1–10 days after NIHL (Fig. 2r), suggesting that PNs with BF across all the tested frequency regions of the cochlea show increased gain after NIHL. Finally, we did not observe a change in either PN threshold (Fig. 2k) or gain (Fig. 2n) in sham-exposed mice (218 neurons from 5 mice, Fig. 2k, n and Supplementary Fig. 2f–j). In sum, whereas the recovery in response threshold was less robust for PNs with high BF, the increase in gain was more uniform across all frequencies. Together, our results show that despite the persistent peripheral damage, A1 L2/3 PNs show recovered response thresholds and response amplitudes and increased response gain.

### The computational model generates testable hypotheses for SOM and PV plasticity after NIHL

Our results showing increased gain and recovered response threshold of PNs after peripheral damage are consistent with previous studies[10,11,14,16,17,20,21]. However, the central goal of our study is to highlight the plasticity of the different cortical IN subtypes and how it may contribute to the observed PN recovery from peripheral damage. Thus, we investigated the possible role of inhibitory circuitry on the recovery of A1 L2/3 PNs after NIHL. To this end, we first used a computational model to investigate the possible changes in inhibition that can achieve PN high gain. Past modeling work has shown that a decrease in the recurrent inhibition in a recurrently coupled cortical model results in higher PN gain[60,61], thus, an NIHL-induced reduction in inhibition could be a candidate mechanism. However, strong recurrent PN connections can yield unstable, run-away behavior if a recurrently coupled inhibitory population is unable to dynamically track and cancel the recurrent excitatory activity[53,60,62]. As such, the stabilization role for inhibition poses constraints on how reduced inhibition can safely modulate cortical gain because a global and non-specific inhibitory reduction could lead to instability and pathology, such as epileptic-like activity[54]. Thus, a simplified two-population model, consisting of generic excitatory and inhibitory neurons, would likely fall short of capturing the experimental results presented thus far[60] (Fig. 2). As a result, we started our investigation by considering a computational network of leaky integrate-and-fire neuron models ("Methods") of three sub-populations of neurons (PNs, PVs, and SOMs) (Fig. 3a). PNs and PVs received a feedforward presynaptic drive, and we modeled sound level by increasing the firing rate of the feedforward inputs. We considered four sound levels: none (no sound), low, medium, and high. The control (pre-exposure) spiking behavior of the network lies in an asynchronous (stable) regime, with the firing rate of all three populations increasing monotonically with sound level (Fig. 3b, c). Because peripheral damage reduces the intensity of peripheral sensory drive from the cochlea to the AN and the brain, noise-induced damage in our model is implemented by decreasing the feedforward (evoked) and background (spontaneous) firing rates. We

modeled recovery after NIHL, as observed 10 days after NIHL (Fig. 2), either as a static depolarization or hyperpolarization of individual cortical neurons. The underlying cause behind these inputs could be due to the intrinsic or synaptic mechanisms that restore neuronal threshold post-NIHL. Consistent with our prediction on the constraints on how reduced inhibition can safely modulate cortical gain, we found that depending on the magnitude and sign of these currents to each subpopulation, the network spiking behavior varied drastically, from oscillatory and unstable, to asynchronous and stable with high gain (Fig. 3d).

Because our major focus is to understand the circuit pathways participating in the recovery of PNs' threshold, high gain, and stability after NIHL, we utilized a mean-field circuit theory, which captures the average neuronal firing rate for each of the sub-populations. This allowed us to perform an extensive parameter sweep (Fig. 3c, d; see "Methods" for additional details). Viable parameter sets that matched our experimental observations of PNs were defined as those that produced stable network dynamics with lower PN response thresholds and higher PN gain than in control (see Methods for additional details). Parameter sets that met these criteria yielded average SOM firing rates that were suppressed compared to the control, thus exhibiting little-to-no SOM recovery after damage (Fig. 3e). However, PV and PN firing rates recovered robustly and similarly (Fig. 3e). These successful parameter sets can be further explored by examining the strength and sign (depolarizing vs. hyperpolarizing) of the recovery currents injected into each of the subpopulations (Fig. 3f). Specifically, while PNs and PVs received depolarizing inputs, SOMs largely received hyperpolarizing inputs. These modeling results lead to a pair of testable hypotheses: (1) PVs will have a matched recovery to that of the PNs, and (2) SOMs will not recover post-NIHL.

### Robust PV sound-evoked activity (recovery) after NIHL

To test the first hypothesis, we first investigated the effect of NIHL on response threshold, amplitude, and gain in A1 L2/3 PVs. To selectively target and image sound-evoked responses from PVs, we injected AAV expressing Cre-dependent GCaMP6f (AAV-Flex-GCaMP6f) into the A1 of PV-Cre mice (Fig. 4a). We first employed in vivo wide-field transcranial imaging of populations of PVs in awake mice (Fig. 4a–f). We found that the response threshold of PVs was increased 1 day after noise exposure (Fig. 4d, magenta). However, 3 days after noise exposure, the PV population response threshold was lower than the response threshold of A1 PNs (Fig. 4d), suggesting that PV response thresholds recover even before the response threshold of PNs. Ten days after NIHL, PV response thresholds remained low and were not different from the PN response thresholds (Fig. 4d). Moreover, we found the reduced sound-evoked response amplitudes of A1 PVs 1 day after NIHL (Fig. 4e, red), which recovered by 10 days after NIHL (Fig. 4e, cyan). Importantly, we found that noise exposure increased the PV gain, which remained elevated for 10 days after NIHL (Fig. 4f). We did not observe a change in either PV response threshold or gain in sham-exposed mice (Supplementary Fig. 3a, b). Together, these results demonstrate that PV recovery matches PN recovery.

Next, we performed longitudinal 2P imaging of A1 L2/3 PVs (Fig. 4g–j). We tracked and included in our analysis 82 PVs from 6 different mice for 10 days after NIHL. Consistent with our wide-field imaging results, PVs displayed recovered response thresholds (Fig. 4k, l) and even surpassed pre-noise-exposure responses 10 days after NIHL amplitudes (Fig. 4m, cyan). Moreover, the gain of individual PVs increased after NIHL and remained increased during the 10 days after NIHL (Fig. 4n–p). Most PVs showed increased response gain after NIHL (day 1: 63/82, day 3: 65/82, and day 10: 61/82) (Fig. 4p insets and supplementary Fig. 3g).

To correlate the peripheral damage to the PV sound-evoked properties, we analyzed the response threshold and gain of PVs as a

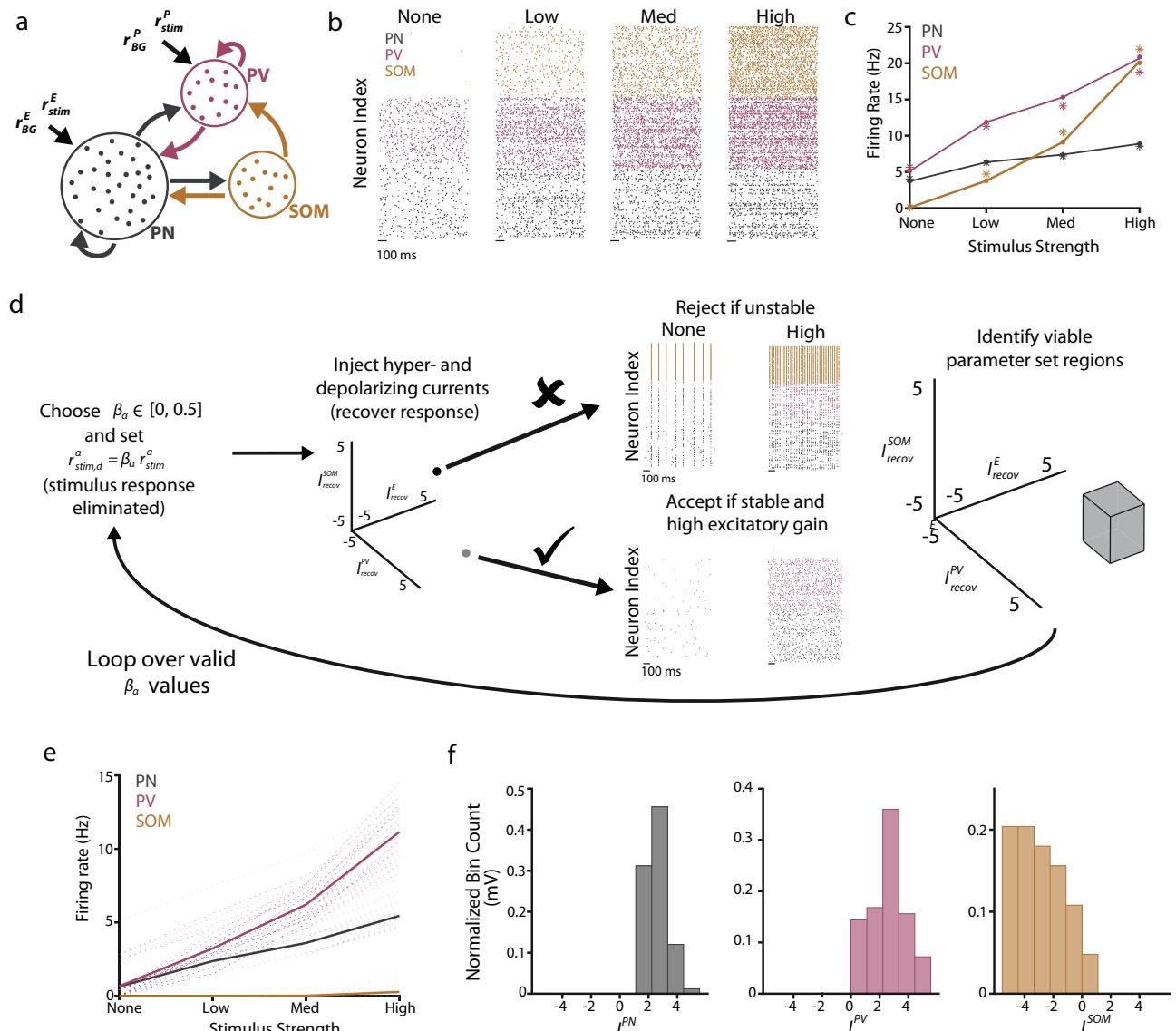

**Fig. 3 | Three-population model generates testable hypotheses for SOM and PV plasticity after NIHL. a** Schematic of connectivity across the three populations (PN, SOM, and PV). **b** Raster plots showing the spiking activity of a subset of neurons at four stimulus levels for the PNs (black), PVs (magenta), and SOMs (orange). **c** Firing rate for the spiking model (dot-line) and mean-field theory (asterisks). **d** Schematic of the parameter sweep algorithm. For specific pairs of damage values ($\beta_{PN,PV}$), the mean-field theory was used to find the firing rates of the model for points in the recovery current space ($I^{PN}_{recov}, I^{PV}_{recov}, I^{SOM}_{recov}$). Parameter sets that yielded stable behavior (asynchronous), along with a low threshold and improved gain (bottom arrow), were accepted, while all others were rejected (e.g., oscillatory, top arrow). Viable parameter regions were identified. This process was looped over all damage values. **e** Firing rate for the three populations. Translucent lines correspond to distinct parameter sets, while bolded lines are the average firing rates across all viable parameter sets. **f** Histograms of the recovery currents were found in the viable parameter sets for the three populations. All parameter values can be found in Supplementary Tables 3 and 4.

function of their BF (Fig. 4q, r). Irrespective of their BF, PVs showed increased response thresholds 1 day after NIHL recovered to pre-exposure levels by 10 days (Fig. 4q). Similarly, we found increased gain in PV neurons with low- to high-BF 1 day after NIHL that remained increased for 10 days (Fig. 4r). These results suggest that PVs with BF across all the frequency regions displayed recovered thresholds and increased gain after NIHL, suggesting that PV recovery is robust and not correlated with the peripheral damage. We did not observe any changes in the response threshold, amplitude, and gain of L2/3 PVs in sham-exposed mice (80 neurons from 7 mice, Fig. 4k, n and supplementary Fig. 3c–g). These results demonstrate that, in response to peripheral damage, A1 L2/3 PVs match the recovery of PNs. Consequently, this recovery is not consistent with PV contribution to the increased PN gain after recovery from NIHL. In fact, this result and our

computational modeling results (Fig. 3e, f) are consistent with the notion that PVs may act as stabilizers of A1 network activity after noise trauma.

**Reduced SOM sound-evoked activity (non-recovery) after NIHL**
We next investigated SOM plasticity after NIHL. We started with in vivo wide-field transcranial imaging (Fig. 5) and found that the response threshold of the A1 SOMs was very high 1 day after NIHL, above 80 dB (Fig. 5c, red). Importantly, response thresholds did not recover and remained significantly higher than PV and PN response thresholds even 10 days after noise exposure (Fig. 5d). Additionally, response amplitudes were reduced and did not fully recover even 10 days after NIHL (Fig. 5e, cyan). Finally, we did not observe any gain changes in SOMs (Fig. 5f). We did not observe a change in the response threshold

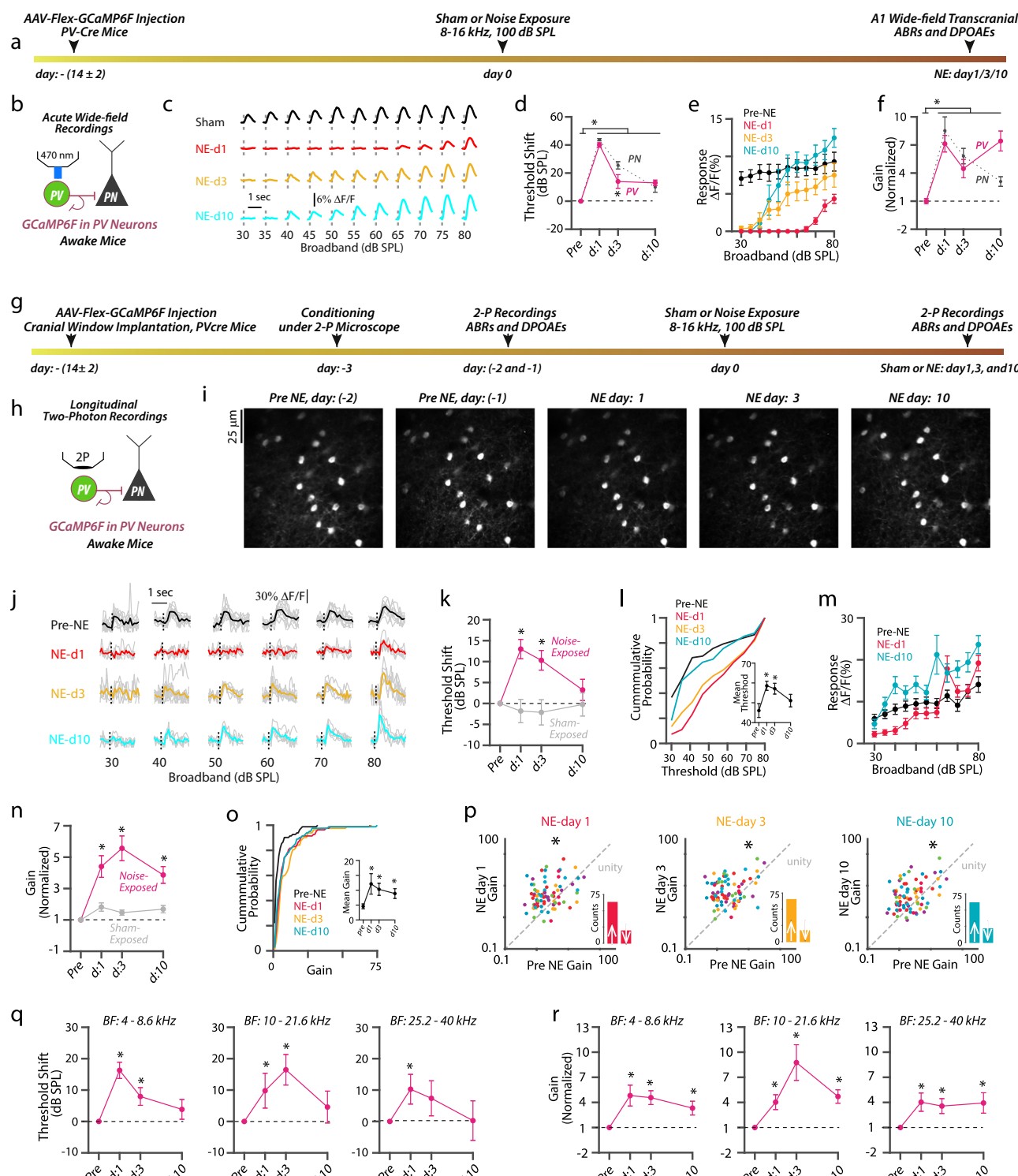

**Fig. 4 | Robust PV sound-evoked activity (recovery) after NIHL. a** Wide-field (WF) imaging experimental design for A1 PVs. **b** Experimental setup. **c** Representative transcranial fluorescence responses of PVs from sham- and noise-exposed mice. **d** Average change in PV response thresholds (magenta) at 1, 3, and 10 days after noise-exposure. The average change in PN threshold (dark gray) reproduced from Fig. 2 (PV vs. PN: effect of cell-type × time, $F = 3.7$, $p = 0.01$). **e** Average sound-evoked responses of A1 PVs to broadband sounds from noise-exposed mice (Intensity vs. time, effect of time, $F = 98.5$, $p < 1.1 \times 10^{-10}$). **f** Average response PV gain (magenta) normalized to pre-noise exposed gain and at 1, 3, and 10 days after noise exposure (effect of exposure, $F = 53.7$, $p = 1.1 \times 10^{-7}$). Normalized PN gain (dark gray) reproduced from Fig. 2. **g** Timetable of longitudinal 2P imaging. **h** Experimental setup illustrating 2P imaging. **i** Z-stack images of tracked PVs. **j** Representative responses from a PV before and after NIHL. **k** Average change in response threshold of individual PV neurons from noise (magenta) and sham (gray) exposed mice (Noise-exposed: 82 neurons from 6 mice, sham-exposed: 80 neurons from 7 mice, noise vs. sham effect of exposure, $F = 11.17$, $p = 0.001$). **l** Cumulative probability of PV response threshold. Inset: Average mean threshold of PVs per mouse. **m** Average responses of individual PVs. **n** Average gain of individual PVs normalized to pre-exposed gain from noise (magenta) and sham (gray) mice. **o** Cumulative probability of PVs gain. Inset: Average mean gain of PVs per mouse. **p** Scatter plots of the gain of individual PVs before and after NIHL. The dotted line represents unity. Insets: Bar graphs representing the number of neurons showing increased gain (↑ above unity) and reduced gain. **q** Average change in response threshold and gain **r** of PVs with Low-BF (left, $n = 39$ PVs), Mid-BF (middle, $n = 24$ PVs), and High-BF (right, $n = 18$ PVs). *$p < 0.05$; statistical values are given in Supplementary Table 5.

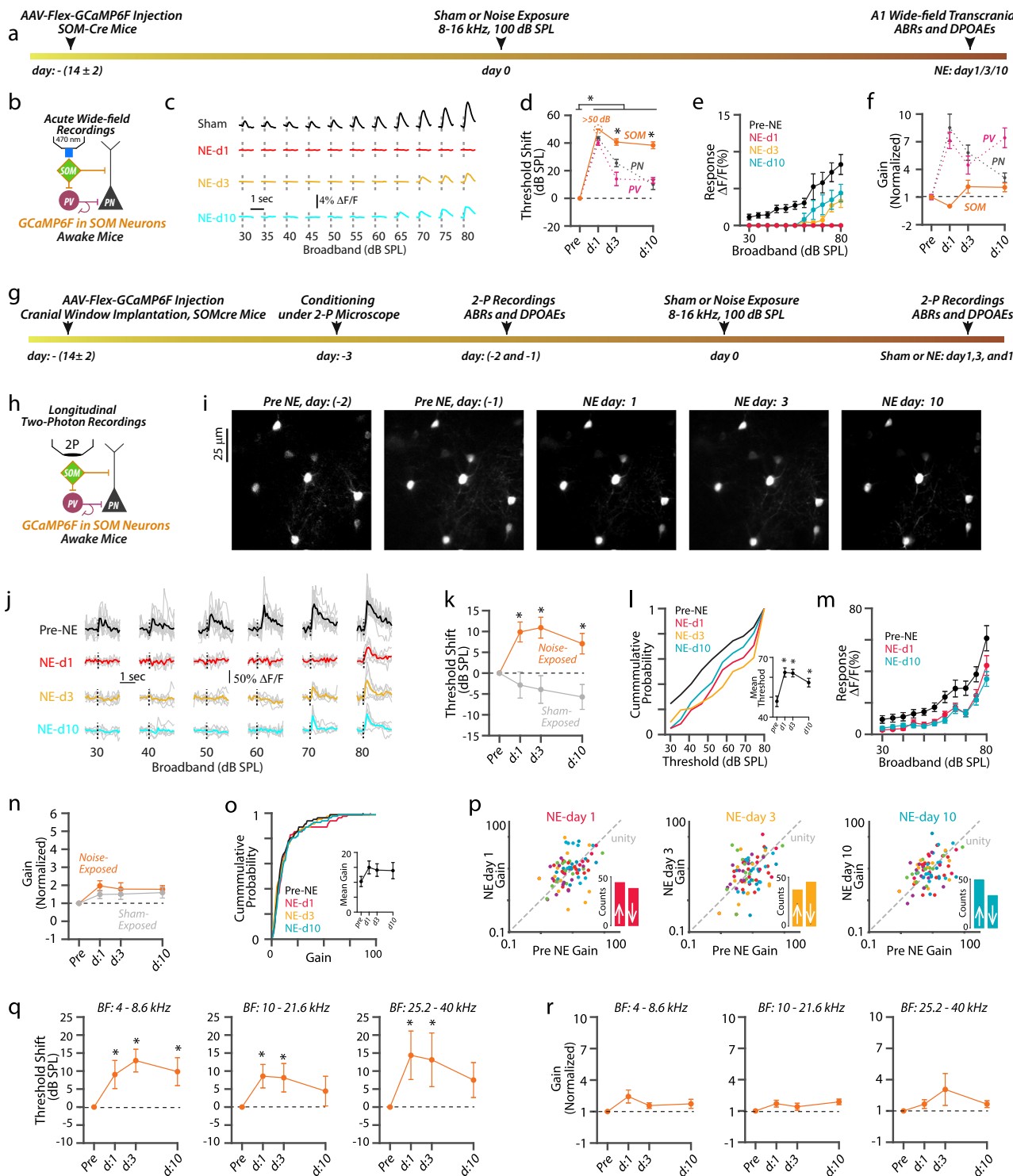

**Fig. 5 | Reduced SOM sound-evoked activity (non-recovery) after NIHL. a** Wide-field (WF) imaging experimental design for A1 SOMs. **b** Experimental setup. **c** Representative transcranial fluorescence responses of A1 SOMs from sham- and noise-exposed mice. **d** Average change in A1 SOM response threshold (orange) before and after noise-exposure. Note: Since we did not observe any sound-evoked activity in SOMs at 1 day after noise exposure, even at 80 dB SPL sounds, we did not assign a threshold shift at this time point, but it is >50 dB SPL. The average change in PN (gray) and PV (red) threshold reproduced from Fig. 4 (PV vs. PN vs. SOM: effect of cell-type, $F = 31.9$, $p = 4.3 \times 10^{-7}$). **e** Average sound-evoked responses of A1 SOMs. (Intensity vs. time, the effect of time, $F = 82.97$, $p < 1.1 \times 10^{-10}$). **f** Average response gain of SOMs (orange) normalized to pre-noise-exposed gain at 1, 3, and 10 days after noise exposure. Normalized PN (gray) and PV (red) gain reproduced from Fig. 4. **g** Timetable of longitudinal 2P imaging. **h** Experimental setup illustrating 2P imaging. **i** Z-stack images of tracked SOMs. **j** Representative responses from a SOM before and

after NIHL. **k** Average change in SOMs response threshold from noise (orange) and sham (gray) exposed mice (Noise vs. sham: effect of exposure, $F = 16.60$, $p = 8.2 \times 10^{-5}$). **l** Cumulative probability of SOM response threshold. Inset: Average mean threshold of SOMs per mouse. **m** Average responses of individual SOMs. **n** Average gain of individual SOMs normalized to pre-exposed gain from noise (orange) and sham (gray) mice. **o** Cumulative probability of SOMs gain. Inset: Average mean gain of SOMs per mouse. **p** Scatter plots of the gain of individual A1 L2/3 SOMs before and after NIHL. The dotted line represents unity. Insets: Bar graphs representing the number of neurons showing increased gain (↑ above unity) and reduced gain (↓ below unity) after NIHL. **q** Average change in response threshold and gain **r** of SOMs with Low-BF (left, $n = 31$ SOMs), Mid-BF (middle, $n = 34$ SOMs), and High-BF (right, $n = 15$ SOMs). *$p < 0.05$; statistical values are given in Supplementary Table 6.

and gain of SOMs in sham-exposed mice (supplementary Fig. 4a b). Overall, in contrast to the robust sound-evoked PN and PV activity after noise trauma, SOM sound-evoked activity remained significantly reduced throughout the 10 days after noise trauma.

Consistent with our wide-field imaging results, longitudinal 2P imaging of individual A1 L2/3 SOMs (82 neurons from 15 mice) showed increased response thresholds after injury, which remained elevated throughout the 10 days after NIHL (Fig. 5g–k) and a shift in the cumulative distribution of response threshold towards higher sound levels (Fig. 5l). Also, consistent with the wide-field imaging results, we found reduced sound-evoked amplitudes of individual SOMs (Fig. 5m, cyan). Moreover, we did not observe any change in the gain of A1 L2/3 SOMs after NIHL (Fig. 5n–p).

When we analyzed SOM response threshold and gain as a function of their BF (Fig. 5q, r), we found that irrespective of their BF, SOMs showed increased response thresholds 1 and 3 days after noise exposure (Fig. 5q). However, 10 days after noise exposure, SOMs with mid- and high-BF showed partial recovery in their response thresholds (Fig. 5q). These results suggest that SOMs with BF corresponding to the high-frequency region of the cochlea, which showed more damage compared to the low-frequency region (Fig. 1, Supplementary Fig. 1a), show partial recovery in their response thresholds after NIHL. Since SOM response threshold plasticity is opposite to PN threshold plasticity [Low-BF: PNs recover but not SOMs−High-BF: SOMs recover but not PNs (Figs. 2q and 5q)], these results are consistent with the notion that SOM threshold plasticity is linked with PN threshold plasticity. In terms of gain, consistent with the overall no change in SOM gain (Fig. 5n), we did not find a change in the gain of SOMs when binned as per their BF (Fig. 5r). Finally, we did not observe a change in SOM response threshold, amplitude, and gain in sham-exposed mice (Fig. 5k, n and supplementary Fig. 4c–g, 42 neurons from 9 mice). In total, these results are consistent with the second modeling prediction that SOM responses are suppressed during recovery from NIHL. Together, these results and our computational modeling results (Fig. 3e, f) are consistent with the notion that the reduced SOM activity disinhibits PVs and PNs, thus allowing for high PV and PN response gain after NIHL.

## SOM intrinsic excitability does not change after NIHL

We next explored the mechanisms underlying suppression in SOM sound-evoked responses after noise trauma. The reduction in SOM activity might be due to changes in the intrinsic cellular excitability of SOMs, the synaptic input afferent to SOMs, or a combination of the two mechanisms. To test for changes in intrinsic properties, we performed ex vivo brain slice electrophysiology in AC L2/3 SOMs after NIHL (Fig. 6). Due to the lack of cytoarchitectural features, it is challenging to locate the AC in brain slices. Therefore, to localize the AC, we labeled AC corticocollicular (CCol) L5B PNs (red) projecting to the inferior colliculus by injecting red fluorescent retrograde microspheres into the inferior colliculus of SOM-GFP mice (Fig. 6a, Methods). The localization of CCol PNs in the AC, along with anatomical landmarks, such as the rhinal fissure and the underlying hippocampal formation, allowed us to locate the AC as described previously[63–65]. After localizing the AC, we measured SOM intrinsic properties in noise- and sham-exposed mice (Fig. 6b). The input resistance ($R_{input}$) and the membrane resting potential ($V_{rest}$) did not change over the 10 days after noise- compared to sham-exposure (Fig. 6c–e). Similarly, noise trauma did not affect action potential width ($AP_{width}$), AP threshold ($AP_{threshold}$), and firing rate (Fig. 6f–k). Finally, the firing rate adaptation ratio, calculated as the ratio of instantaneous firing frequency between the ninth and tenth AP and instantaneous frequency between the second and third AP (f9/f2)[64], showed no significant difference between sham- vs. noise-exposed mice (Fig. 6l, m). Taken together, these results suggest that the reduced sound-evoked activity in SOMs after NIHL is likely not due to changes in SOM intrinsic properties.

## The computational model generates testable hypotheses for VIP plasticity after NIHL

We next investigated whether changes in the synaptic inputs to SOMs are associated with SOM plasticity after NIHL. Although our three-population model correctly predicted a cell-type-specific suppression of SOMs, it cannot capture such a synaptic mechanism in its current form: SOMs lack significant recurrent inhibition from either themselves or PVs[35,36,66] (Fig. 3a). However, VIPs, which were not included in our initial model, are strongly embedded in the AC recurrent network. They have substantial incoming connections from PNs, PVs, and SOMs, and considerable outgoing connections onto SOMs[66,67]. Most notably, the strong mutual inhibition between SOMs and VIPs (Fig. 7a; highlighted) potentially drives a competitive dynamic between these two subpopulations, where tipping the activity in favor of one subpopulation could lead to a dramatic suppression of the other subpopulation[68]. We, therefore, extend our computational model to include VIPs to investigate their plasticity and its potential association with the observed SOM suppression.

For the control (pre-damaged) state, we found that the four-population model exhibited similar spiking behavior as in the three-population model (Fig. 7b) and that the mean-field theory was readily extendable to accurately capture the underlying steady-state firing rates (Fig. 7c). After establishing this baseline spiking behavior, we next performed a similar parameter sweep as before (see "Methods" for additional details). We found that the firing rates corresponding to the viable parameter sets (i.e., parameter values that yielded a low threshold, high gain, and stable dynamics for the PNs) for the PN, PV, and SOM neuronal subpopulations in the extended four-population model matched those found in our simplified three-population model (Fig. 3e). Specifically, the population-averaged firing rates of PNs and PVs showed low threshold and high gain, while the SOMs were largely suppressed (Fig. 7d). The difference here was that the inhibition of SOMs was brought on solely by VIPs and not by a hyperpolarizing recovery current (as was the case in the three-population model in Fig. 3). In line with this observation, VIPs exhibited an increase in firing rates after damage compared to control, while also showing similar characteristics as the PNs and PVs, namely a low threshold and high gain (Fig. 7d). After examining the recovery currents responsible for these results, we observed that PNs, PVs, and VIPs were all subjected to significant depolarizing currents, with VIPs receiving the strongest level of depolarization (Fig. 7e). This result combined with the strong VIP to SOM connection suggests that, during recovery after trauma, SOMs are more inhibited compared to the control state. To test this directly, we measured the average synaptic input to SOMs for all viable parameter sets (Fig. 7f, see Methods for additional details). We found that the average synaptic input was less after trauma when compared to control across all viable parameter sets and stimulus values (Fig. 7f). Further, for a majority (51.47%) of these tested conditions, SOMs received a net inhibitory input. In our four-population model (Fig. 7a), to simplify the parameter space, we included a direct, stimulus-dependent excitatory input onto VIPs. Since there is no evidence so far that VIPs receive direct thalamic input[69], this input could be viewed as a simplification of feedback excitatory inputs from higher-order processing, deeper cortical layers, and short-term facilitating connections from recurrently connected PNs[70–72]. Removing the direct stimulus input and allowing the strength of excitatory input onto VIPs to vary with stimulus strength produces model results that are quantitively very similar to those where VIPs receive direct stimulus inputs (Supplementary Fig. 5 and see below on the role of intrinsic VIP plasticity). In total, these modeling results from the four-population model provide a clear, testable hypothesis: VIP neurons show a strong recovery after noise trauma.

## Robust VIP sound-evoked activity (recovery) after NIHL

To test this hypothesis experimentally, we first used in vivo wide-field transcranial imaging of populations of VIPs (Fig. 8a–f). We found that

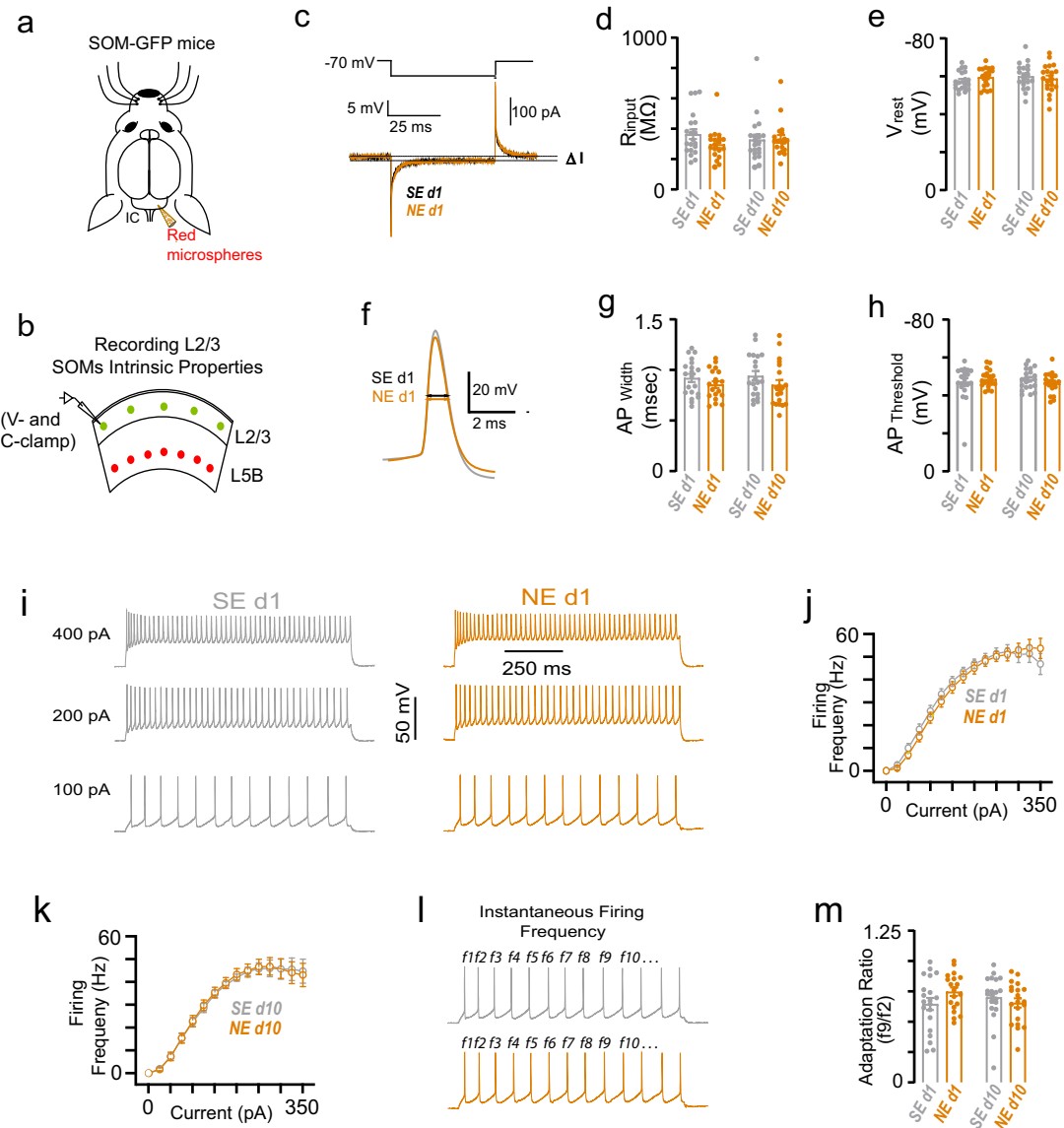

**Fig. 6 | SOM intrinsic excitability does not change after NIHL. a** Schematic illustration of stereotaxic injections of red retrograde microspheres into the right IC to label CCols and identify the AC in the brain slices. **b** Schematic illustration of brain slice electrophysiology experiment showing recordings of AC L2/3 SOMs. Red circles represent the L5B CCols. Green circles represent SOMs. **c** Schematic of hyperpolarizing pulses (top) and representative transient current (bottom) responses in SOM neurons in voltage–clamp recording mode. **d** Average input resistance ($R_{inp}$) of L2/3 SOM neurons after noise- or sham-exposure. Filled circles represent the $R_{inp}$ of individual SOMs (SEday 1: 20 neurons from 3 mice, NEday 1: 19 neurons from 3 mice, SEday 10: 20 neurons from 3 mice, and NEday 10: 20 neurons from 3 mice; effect of exposure, $F = 0.89$, $p = 0.34$). **e** Average SOM resting membrane potential after noise- or sham-exposure. Filled circles represent the resting membrane potential of individual SOMs (effect of exposure, $F = 0.07$, $p = 0.78$). **f** Representative action potential (AP) waveforms. Arrows indicate AP width.

**g** Average AP width of SOMs neurons after noise- or sham-exposure. Filled circles represent the AP width of individual SOM neurons (effect of exposure, $F = 2.6$, $p = 0.11$). **h** Average SOM AP threshold after noise- or sham-exposure. Filled circles represent the AP threshold of individual SOMs. (effect of exposure, $F = 0.02$, $p = 0.86$). **i** Representative SOM firing in response to increasing depolarizing current (100, 200, 400 pA current injections) 1 day after sham (gray) and noise (orange) exposure. **j** Average firing frequency of SOMs as a function of injected current amplitude 1 day after sham (gray) and noise (orange) exposure. **k** Same as **j** but 10 days after sham (gray) and noise (orange) exposure. **l** Temporal pattern of action potential generation of SOMs after sham (gray) and noise (orange). **m** Average SOM adaptation ratio (f9/f2, see panel **l** for traces) rate after noise- or sham-exposure. Filled circles represent the adaptation ratio of individual SOMs (effect of exposure, $F = 0.42$, $p = 0.51$). Statistical values are given in Supplementary Table 7.

the response threshold of A1 VIPs was significantly lower than the response threshold of PNs, PVs, and SOMs 1 day after NIHL and showed full recovery by 10 days after NIHL (Fig. 8d, cyan). Further, VIP response amplitudes surpassed their pre-noise-exposure amplitudes 10 days after NIHL (Fig. 8e, cyan), and the gain was also increased throughout the 10 days after NIHL (Fig. 8f). We did not observe a change in either VIP response threshold or gain in sham-exposed mice (supplementary Fig. 6a, b). Consistent with our transcranial results, longitudinal 2P imaging of individual A1 L2/3 VIPs also revealed recovered (low) response thresholds, robust and even enhanced response amplitudes, and increased gain (Fig. 8g–p, 70 neurons from 8 mice). Also, most VIPs showed increased gain after NIHL (Fig. 8p insets and Supplementary Fig. 6g; day 1: 36/66, day 3: 43/70, and day 70: 47/70). On the other hand, we observed slightly reduced VIP gain and no change in either VIP response threshold or amplitude VIPs in sham-exposed mice (60 neurons from 6 mice, Fig. 8k, n and Supplementary Fig. 6c–g), suggesting that we may have underestimated the increase in VIP gain after NIHL.

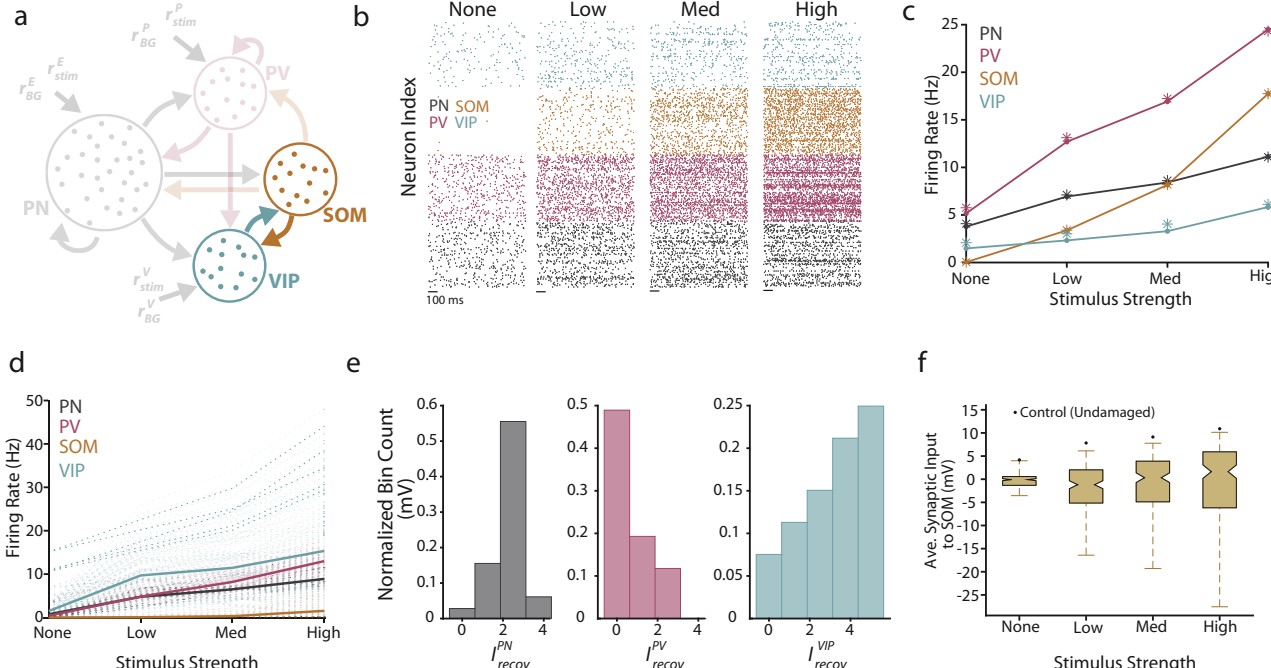

**Fig. 7 | Four-population model generates a testable hypothesis for VIP plasticity after NIHL. a** Schematic of the four-population model, with the mutual inhibition between VIP and SOM neurons highlighted. **b** Raster plots showing the spiking activity of a subset of neurons at four stimulus levels for the PN (black), PV (magenta), SOM (orange), and VIP (cyan) populations. **c** Firing rate for the spiking model (dot-line) and mean-field theory (asterisks). **d** Firing rate for the four populations. Translucent lines correspond to distinct parameter sets, while bolded lines are the average firing rates across all viable parameter sets. **e** Histograms of the

recovery currents were found in the viable parameter sets for the PN, PV, and VIP populations. SOMs did not receive any direct recovery current in this parameter search. **f** Box plots showing the range of average synaptic input to SOMs for the viable parameter sets ($n = 170$), along with the value for the default (sham) case (black dot). Box plots show the minimum (lower whisker), first quartile (lower bound of the box), median (box center), third quartile (upper bound of the box), and maximum (upper whisker). All parameter values can be found in Supplementary Tables 3 and 4.

When we analyzed VIP response threshold and gain as a function of their BF (Fig. 8q, r), we found that VIPs with low- and mid-BF showed increased response thresholds 1 day after NIHL, which were recovered to baseline levels by 10 days after NIHL (Fig. 8q). However, VIPs with high-BF showed reduced recovery in their response thresholds (Fig. 8q). These results suggest that VIPs with BF corresponding to the high-frequency region of the cochlea, which showed more damage compared to the low-frequency region (Fig. 1 supplementary Fig. 1a), do not recover their response thresholds completely. In terms of gain, VIPs showed increased gain after NIHL across all the frequencies (Fig. 8r). Taken together, our results support a strong recovery of VIP activity after noise trauma, even surpassing the control activity. Because VIPs inhibit SOMs[35] (Fig. 8b), these results are consistent with the circuit mechanism where robust VIP activity enables SOM suppression, which in turn leads to high PN and PV gain.

### Increased VIP intrinsic excitability after NIHL

To explore the mechanism underlying the enhanced VIP activity after noise trauma, we compared VIP intrinsic excitability between sham- and noise-exposed mice at 10 days after trauma. Namely, after localizing the AC (Fig. 9ab, Methods), we performed brain slice electrophysiology experiments in AC L2/3 VIPs to assess their intrinsic excitability (Fig. 9, Methods). Although the resting membrane potential, AP width and threshold, and adaptation ratio did not change, we found increased input resistance (Fig. 9d) and firing rate (Fig. 9j) in VIPs from noise-exposed mice. Although we can't exclude the potential contributions of synaptic changes, this result indicates increased VIP neuron excitability 10 days after NIHL and is consistent with the notion that this increased intrinsic excitability might contribute to the enhanced recovery of VIP sound-evoked responses.

## Discussion

### Cell-type-specific plasticity of cortical IN subtypes after NIHL: strengths and limitations of our approach

Extensive evidence supports divergence, complementarity, and division-of-labor between the cortical IN subtypes in terms of their tuning properties[41,73–75] and their role in contextual and adaptive cortical sound processing[43,46,76,77]. However, despite the established role of reduced GABAergic signaling in A1 plasticity after cochlear damage[11,21,27,28,78,79], the plasticity of different IN subtypes in cortical recovery remained unknown. Our approach with computational modeling, in vivo imaging, and in vitro electrophysiology provides a comprehensive account of the distinct plasticity of the different cortical IN subtypes during cortical recovery. This distinct, cell-type-specific plasticity supports the notion that PVs, SOMs, and VIPs play distinct roles in cortical recovery after peripheral damage.

Namely, after noise trauma, we found enhanced sound-evoked activity in PNs (Fig. 2), PVs (Fig. 4), and VIPs (Fig. 8) but reduced sound-evoked activity in SOMs (Fig. 5). Based on the known sequentially organized inhibitory cortical network[35], where VIPs neurons inhibit SOMs, SOMs inhibit PVs and PNs, and PVs inhibit other PVs and PNs, our results are consistent with the notion that the underlying SOM → PV → PN and SOM → PN circuits support a cell-type-specific plasticity mechanism in which, robust PV activity provides network stability by balancing PN activity. Moreover, the vastly decreased SOM activity allows for increased PV and PN gain, which supports stability and high gain. The VIP → SOM → PN disinhibitory pathway completes the task, whereby robust VIP activity enables reduced SOM activity. Although the causality or even the contribution of the proposed roles of PVs, SOMs, and VIPs and recovered A1 sound processing were not tested here, they will be evaluated in future work. Moreover, we are aware

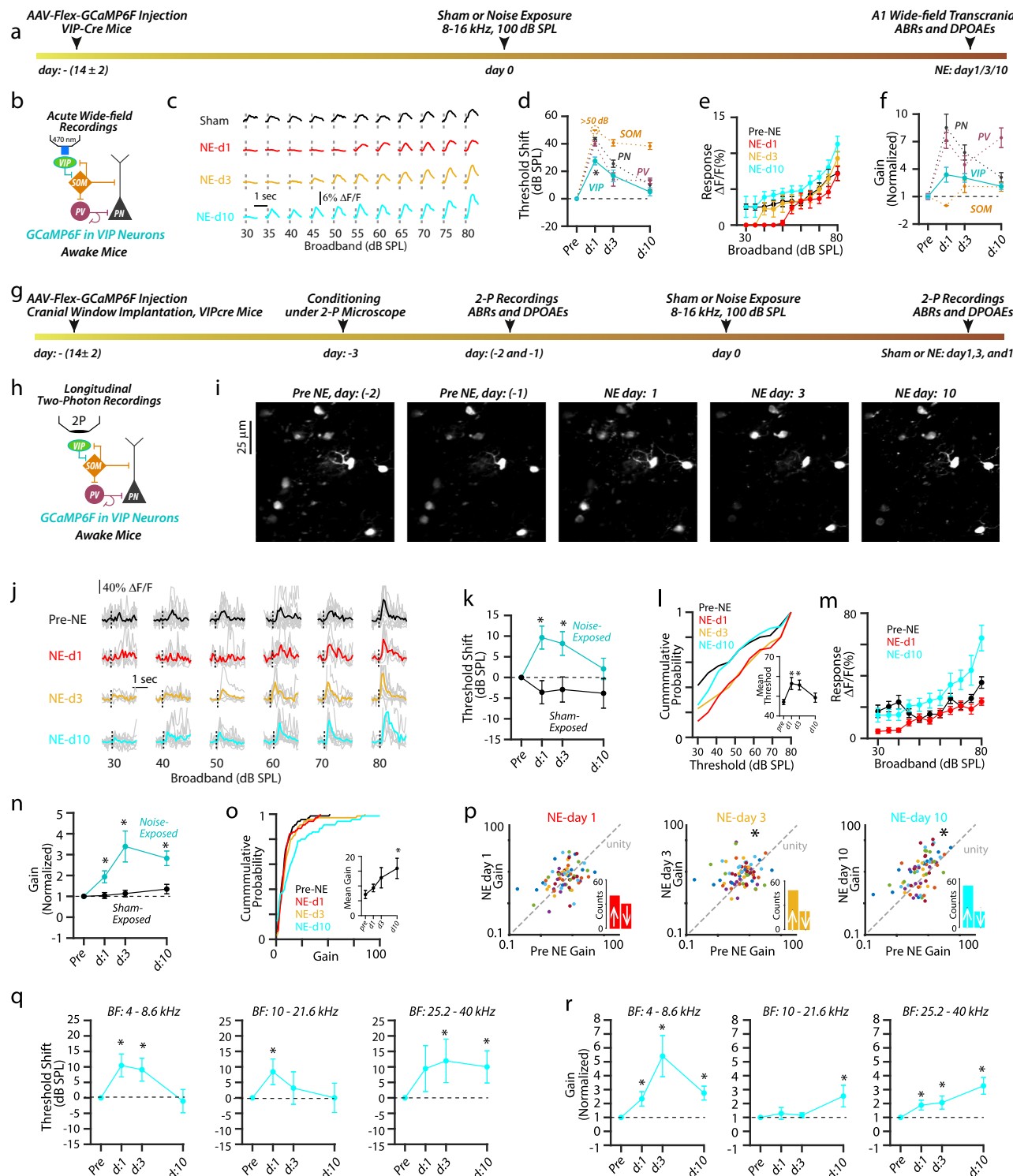

**Fig. 8 | Robust VIP sound-evoked activity (recovery) after NIHL. a** Wide-field (WF) imaging experimental design for A1 VIPs. **b** Experimental setup.
**c** Representative transcranial fluorescence responses of A1 VIPs from sham- and noise-exposed mice. **d** Average change in A1 VIP response thresholds (cyan) before- and at 1, 3, and 10 days after noise exposure. The average change in PN (gray), PV (red), and SOM (orange) thresholds reproduced from Fig. 5 (PV vs. PN vs. SOM vs. VIP: effect of cell-type, $F = 32.8$, $p = 5.4 \times 10^{-9}$). **e** Average sound-evoked responses of A1 VIPs from noise-exposed mice (intensity vs. time, the effect of time, $F = 29.66$, $p < 1.1 \times 10^{-10}$). **f** Average response gain of VIPs (cyan), normalized to pre-noise-exposed gain before and after noise exposure. (effect of exposure, $F = 13.51$, $p = 0.007$). Normalized PN (gray), PV (magenta), and SOM (orange) gain reproduced from Fig. 5. **g** Timetable of longitudinal 2P imaging experimental design for A1 L2/3 VIPs. **h** Experimental setup illustrating longitudinal 2P imaging. **i** Z-stack images of tracked VIPs. **j** Representative responses from VIPs before and after NIHL.

**k** Average change in response threshold of individual VIPs from noise (cyan) and sham (gray) exposed mice (Noise-exposed: 70 neurons from 8 mice, sham-exposed: 60 neurons from 6 mice, noise vs. sham: effect of exposure, $F = 9.2$, $p = 0.002$). **l** Cumulative probability of VIPs response threshold. Inset: Average mean VIP threshold per mouse. **m** Average responses of individual VIPs. **n** Average gain of individual VIPs normalized to pre-exposed gain from noise (cyan) and sham (gray) mice. **o** Cumulative probability of VIP gain. Inset: Average mean gain of VIPs per mouse. **p** Scatter plots of the gain of individual VIPs before and after NIHL. Dotted line represents unity. Insets: Bar graphs represent the number of neurons showing increased gain (↑ above unity) and reduced gain (↓ below unity) after NIHL. **q** Average change in response threshold and **r** of VIPs with Low-BF (left, $n = 33$ VIPs), Mid-BF (middle, $n = 18$ VIPs), and High-BF (right, $n = 17$ VIPs). *$p < 0.05$; statistical values are given in Supplementary Table 8.

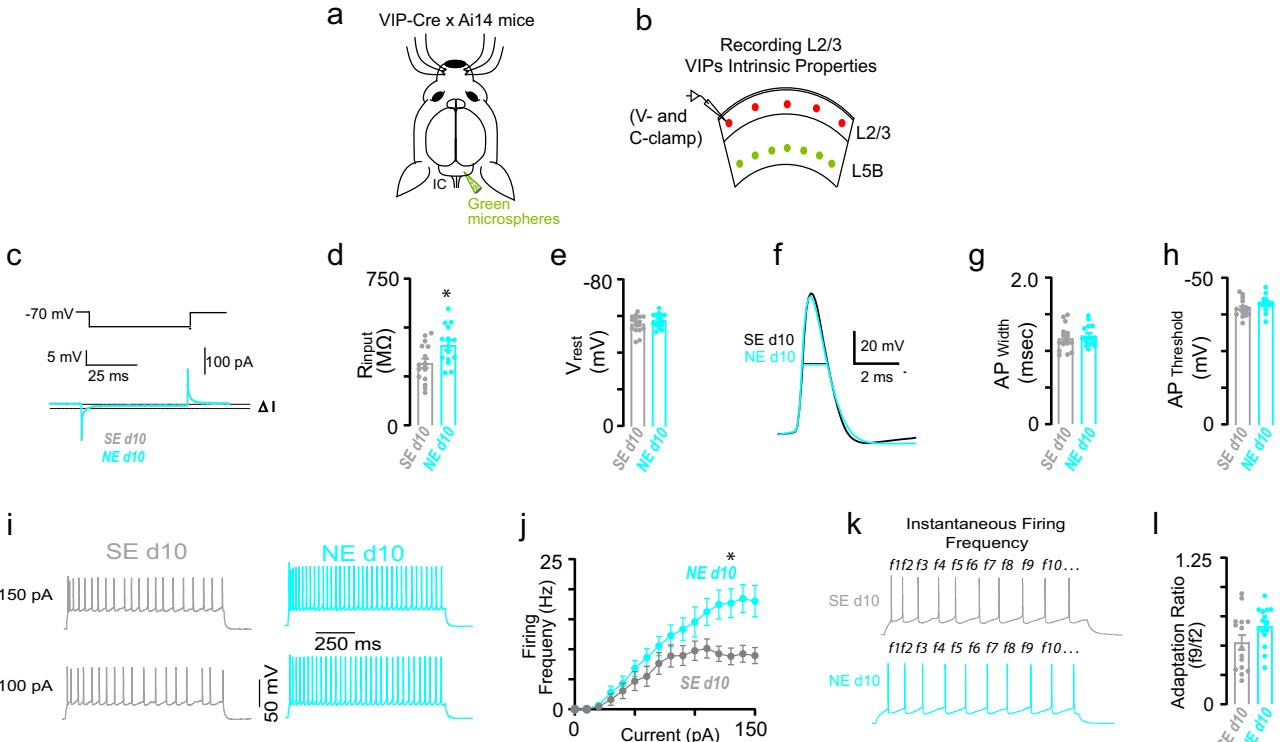

**Fig. 9 | Increased VIP intrinsic excitability after NIHL. a** Schematic illustration of stereotaxic injections of red retrograde microspheres into the right inferior colliculus to label CCols and identify the AC in the brain slices. **b** Schematic illustration of brain slice electrophysiology experiment showing recordings from AC L2/3 VIPs. Green circles represent Ccols. Red circles represent the L2/3 VIPs. **c** Schematic of hyperpolarizing pulses (top) and representative transient current (bottom) from VIP responses in voltage-clamp recording mode. **d** Average VIP input resistance ($R_{inp}$) after noise- or sham-exposure. Filled circles represent the $R_{inp}$ of individual VIPs (unpaired $t$-test, $p = 0.01$). **e** The average resting membrane potential of VIPs neurons after noise- or sham-exposure. Filled circles represent the resting membrane potential of individual VIPs (Mann–Whitney test, $p = 0.31$). **f** Representative action potential (AP) waveforms. Arrows indicate AP width. **g** Average VIP AP width after noise- or sham-exposure. Filled circles represent the AP width of individual VIPs (Mann–Whitney test, $p = 0.51$). **h** Average VIP AP threshold after noise- or sham-exposure. Filled circles represent the AP threshold of individual VIP neurons (Mann–Whitney test, $p = 0.07$). **i** Representing firing of VIPs in response to increasing depolarizing current (100, 150 pA current injections) 10 days after sham (gray) and noise (cyan) exposure. **j** Average VIP firing frequency as a function of injected current amplitude 10 days after sham (gray) and noise (cyan) exposure (2-way ANOVA, effect of exposure: $F = 46.7$, $p = 3 \times 10^{-10}$). **k** Temporal pattern of action potential generation in VIPs after sham (gray) and noise (cyan). **l** Average adaptation ratio (f9/f2, see panel **k** for traces) of VIP firing rate after noise- or sham-exposure. Filled circles represent the adaptation ratio of individual VIPs (Mann–Whitney test, $p = 0.10$). $n = 16$ neurons from 3 mice for SEday 10 and 15 neurons from 3 mice for NEday 10. Group data are presented as mean ± SEM. All the statistical tests were two-sided.

that the recovery of cortical sound processing after noise trauma is influenced by subcortical plasticity mechanisms[10,32], which are also outside the scope of this study. In sum, our study is the first to identify the cell-type-specific plasticity of the different cortical IN subtypes in cortical recovery after noise trauma; and it suggests distinct roles for these interneurons in this process. As such, these results highlight a strategic, cooperative, and cell-type-specific plasticity program that may contribute to restoring cortical sound processing after cochlear damage and, thus, may provide novel cellular targets that may also aid in the development of pharmacotherapeutic or rehabilitative treatment options for impaired hearing after NIHL.

Several key cortical circuit features are consistent with our proposed hypothesis regarding the proposed roles of PVs as stabilizers and SOMs as modulators of A1 plasticity after NIHL. PNs and PVs are embedded into very similar synaptic environments. Both receive the excitatory drive from upstream areas[35,36], and both receive strong recurrent excitation, as well as PV- and SOM-mediated inhibition[66]. This symmetry places PVs in a strategic position for monitoring and stabilizing PN activity[60]. On the other hand, SOMs are in a better position to modulate the cortical inhibition and excitation, such that a higher gain state of PNs can be achieved without compromising the stability of the network. For example, the increased PN gain via reduced SOM activity would also lead to increased PV firing because of

the strong excitatory feedback from PNs to PVs. In turn, these hyperactive PVs would then stabilize the recurrently activated PNs. Further, consistent with our model predictions (Fig. 3e), suppression of SOMs enhances cortical plasticity without compromising the stability of the network[54,80,81], whereas suppression of PVs can result in uncontrolled network activity, evidenced by unstable ictal-like events in most[54], but not all cases[82].

Although we do not present a measure of network stability, future studies could utilize optogenetic perturbations to further investigate our proposed roles of PVs, SOMs, and VIPs while also evaluating how network stability changes after NIHL. Namely, different light intensities could be utilized to suppress a varying fraction of PVs, SOMs, and VIPs. Runaway excitation would arise while targeting the subpopulation responsible for stabilizing the network. Moreover, the light intensity at which runaway activity occurs can be used as a proxy for network stability, with higher light intensities indicating greater stability. To assess how the network stability changes during recovery, the same experiment can be repeated before and at different times after NIHL. Taken together, our observations are consistent with the notion, albeit not tested here, that PVs act as the stabilizers, whereas SOMs act as the modulators of A1 plasticity.

Similarly, our proposed role of VIPs as the enablers of A1 plasticity after NIHL is consistent with previous reports showing

that VIPs enable cortical plasticity across the sensory cortices[68,77,80,81,83–85]. In the visual cortex, synaptic transmission from VIPs to SOMs is necessary and sufficient for the increased cortical responses in PNs after monocular deprivation in adult mice[80]. In the somatosensory cortex, increased VIP activity facilitates increased PN activity in a mouse model of neuropathic pain[86]. Here, we found the enhanced activity of A1 L2/3 VIPs after NIHL (Fig. 8), suggesting that VIPs may contribute to the enhanced PN activity (Fig. 2) via the disinhibitory pathway VIP → SOM → PN. Together, our results provide the first comprehensive cell-type- and circuit-plasticity mechanism that may explain, at least partially, how cortex rebuilds itself after peripheral damage.

## Maladaptive aspects of cortical plasticity

Compensatory A1 plasticity after peripheral damage likely supports the recovery of the perceptual sound-detection threshold but does not support sound processing encoded by precise spike timing, such as modulated noise or speech, and restricts hearing in a noisy environment[10,11,15,87,88]. Interestingly, A1 SOMs, which are critically important for sound processing encoded by precise spike timing of neuronal firing[38,46,89,90], showed reduced sound-evoked activity after NIHL (Fig. 5). Based on these results and although not tested here, we propose that the reduced SOM activity after cochlear damage may contribute to the hearing problems after peripheral damage, such as difficulty in understanding speech and trouble hearing in noisy environments.

Another maladaptive aspect of increased AC gain after peripheral damage is the development of tinnitus, the perception of phantom sounds[15], and hyperacusis, the painful sensitivity to everyday sounds[16]. Both these disorders have many similarities with neuropathic pain and phantom limb syndrome[91]. Namely, these neurological disorders are developed after damage to the peripheral organs and manifest increased PN activity in the respective sensory cortices, suggesting a common underlying cortical circuit mechanism. In the allodynia mouse model of neuropathic pain[86], where sensory touch that does not normally provoke pain becomes painful in the spared nerve injury model, SOM activity was drastically reduced in the somatosensory cortex. This is consistent with the notion that reduced SOM activity disinhibits PNs and leads to increased activity of PNs. Interestingly, selective SOM activation in the somatosensory cortex after nerve injury was sufficient to prevent the increased PN activity and mitigate the development of neuropathic pain[86]. Together, these results suggest that the reduced cortical SOM activity after peripheral organ damage may be a common mechanism across sensory cortices that permits the increased PN gain. Moreover, these results suggest that the modulation of A1 SOM activity after noise trauma could be a potential target for mitigating noise-induced tinnitus and hyperacusis.

## PVs and cortical plasticity after peripheral trauma

Our results suggest that the increased PV sound-evoked activity 10 days after NIHL (Fig. 4d, k, cyan) may stabilize the increased PN activity (Fig. 2). However, 1 day after NIHL, we observed reduced sound-evoked PV activity (Fig. 4d, k, red). Since PVs initiate cortical plasticity in juvenile and adult brain[83] an initial reduction in PV activity after NIHL may initiate cortical plasticity. Consistent with this notion, a rapid drop in PV-mediated PN inhibition as early as 1 day after cochlear denervation precedes the recovery of cortical sound processing[10]. Moreover, recent results show reduced intrinsic excitability of AC L2/3 PVs 1 day after noise-trauma because of increased KCNQ potassium channel activity[31], suggesting that this mechanism might contribute to decreased PV responses to sound 1 day after noise exposure. Similarly, in the visual cortex, PVs show reduced firing rates 1 day after monocular deprivation[83], and in the somatosensory cortex, PVs also show a

reduction in their intrinsic excitability 1 day after whisker plugging[92]. These results suggest that a rapid reduction in PV-mediated PN inhibition may be a common feature of sensory cortices plasticity that plays a critical role in initiating cortical recovery after sensory organ damage.

A recent study[11] reported that the PV sound-evoked activity was reduced after cochlear denervation and remained reduced for two weeks. However, unlike our model of noise-induced cochlear damage, cochlear denervation was induced with bilateral cochlear application of ouabain, which eliminates ~95% of the type 1 SGNs. Since the type and the severity of peripheral organ damage may result in heterogeneous cortical plasticity[7,10,16,21,30,93–99], noise- and ouabain-induced damage to the cochlea may trigger different trajectories of plasticity in A1 neuronal subtypes. Another explanation for the observed differences could arise from differences in the experimental design, such as the sound-stimuli used (broadband vs. 12 kHz pure tones[11]) and the number of PVs tracked (82 vs. 29[11]). Overall, the observed differences point to the need for additional rigorous investigations on the role of distinct INs in A1 plasticity in different types and degrees of hearing loss, including unilateral vs. bilateral and noise vs. ototoxic compounds-induced hearing loss. Nonetheless, our results provide a comprehensive model of cortical rehabilitation after noise trauma whereby the precise and well-timed division-of-labor and cooperativity among cortical INs may secure high gain and stability.

## Strengths and limitations of our wide-field and 2P imaging studies

We employed A1 wide-field transcranial imaging and A1 longitudinal 2P imaging of L2/3 somata to identify the plasticity in the different neuronal cell types of A1 after NIHL. Although we found overall similar plasticity rules via 2P and wide-field imaging after NIHL, wide-field imaging showed larger shifts and reduced recovery in the response thresholds compared to the 2P imaging. The main reason that likely explains these differences is that wide-field imaging assesses responses from different neuronal compartments (somata and dendrites) and different cortical layers, whereas 2P imaging assesses only L2/3 somatic responses[56]. As such, wide-field imaging results might provide insights into the cell-type-specific plasticity of the entire A1 after NIHL. For example, a robust shift and lower recovery in the wide-field response thresholds after NIHL is consistent with the notion that A1 dendrites and/or neurons residing in the cortical layers other than L2/3 may undergo more robust and longer-lasting changes than changes occurring in L2/3 somata after NIHL. In this context, one limitation of our study is that we studied cellular mechanisms based on distinguishing PNs, PVs, SOMs, and VIPs; however, these neuronal subtypes are not a homogeneous population[35,36]. Moreover, our 2P studies were only limited to L2/3. Therefore, our future studies will expand to investigate different cortical neuronal subtypes in terms of their gene expression profile, projection targets, and AC layer localization.

## Model assumptions and limitations

In this work, we leveraged a computational model to assist the exploration of cortical mechanisms responsible for the A1 PN recovery following NIHL. At the neuronal level, we modeled recovery as either depolarization or hyperpolarization of the resting membrane potential induced by current injection. After peripheral injuries, such as in NIHL, it is possible that intrinsic or synaptic mechanisms of homeostatic plasticity could lead to such changes in the membrane potential. However, our results did not show a change in resting membrane potential after NIHL. Nonetheless, our modeling studies provided testable hypotheses that facilitated the experimental design.

The computational model assumes that after NIHL, the network must balance recovery and stabilization. The assumption that cortical inhibition is needed to prevent pathologic activity places the model in the inhibition-stabilized network (ISN) regime[46,53,60,62].

While we have already mentioned the hypothesis that PVs are the best IN subtype to stabilize the network, after NIHL, synaptic plasticity, and other mechanisms may divert this role to alternative INs or shift the entire circuit out of the ISN regime. In that case, the region of viable parameter sets for the computational model would grow and, as a result, would suggest alternative recovery pathways (e.g., hyperpolarizing PVs). Yet, the absence of such pathways in our experimental results implies that the dynamics observed in the ISN regime constrain the mechanisms utilized in recovery. However, it remains an open question whether the cortex lies in the ISN regime with inhibition actively preventing unstable dynamics by balancing excitation or is simply wired to resist instability. The answer likely resides in the dynamics and relative strengths of synaptic connections. As a result, future models should account for possible heterogeneities of synaptic strengths across and within populations, as well as the diversity of short-term plasticity rules found across different cortical synaptic connections[72]. Therefore, combinations of computational modeling with ongoing and future ex vivo and in vivo studies in our laboratories are expected to address the synaptic mechanisms underlying cortical rehabilitation after noise trauma. However, it is also clear that in addition to synaptic mechanisms, the cell-type-specific intrinsic plasticity found in PVs[31] and VIPs (Fig. 9), but not in either PNs[31] or SOMs (Fig.5), are consistent with contributions of intrinsic plasticity mechanisms in cortical recovery.

In conclusion, our results create a comprehensive framework for understanding the cellular and circuit mechanisms underlying AC plasticity after peripheral trauma and hold the promise to advance understanding of the cortical mechanisms underlying disorders associated with maladaptive cortical plasticity after peripheral damage, such as tinnitus[14,15], hyperacusis[16,100], visual hallucinations, and phantom limb pain[13,101].

## Methods

### Animals

All procedures were approved by the Institutional Animal Care and Use Committee at the University of Pittsburgh. Mice were housed in a 12 h light/12 h dark cycle with a room temperature of 65–75 °F with 40–50% humidity. For experiments shown in Fig. 1 describing the animal model of noise-induced peripheral damage, male and female PV-Cre, SOM-Cre mice, and VIP-Cre mice with C57/B6 mice backgrounds (The Jackson Laboratory) were noise-exposed at P63–70. ABR, DPOAE, and histology were performed at P62-84. For widefield and 2P experiments shown in Figs. 2, 4, 5, and 8, male and female PV-Cre, SOM-Cre, and VIP-Cre mice were injected with AAVs at P49–56, noise-exposed at P63–70, and imaging was performed at P60-80. For in vitro electrophysiological experiments assessing intrinsic properties in SOM neurons (SOMs) shown in Fig. 6, male and female SOM-GFP (GIN) mice with C57/B6 mice backgrounds were injected with retrograde beads at P28–35, noise-exposed at P49-56, and electrophysiology was performed at P50–66. For in vitro electrophysiological experiments assessing intrinsic properties in VIP neurons (VIPs) shown in Fig. 9, male and female offspring from VIP-Cre x Ai14 (The Jackson Laboratory) were injected with retrograde beads at P28–35, noise-exposed at P49–56, and electrophysiology was performed at P50–66.

### Speaker calibration

Acoustic sound stimuli used in the study were calibrated with pre-amp attached microphones (1/8 in 4138-A-015 and 1/4 inch 4954-B, Brüel and Kjær) and a reference 1 kHz, 94 dB SPL certified speaker (Type 4231, Bruel & Kjaer). More specifically, we placed the microphone at the same position as the mouse ear and delivered the pure tones and broadband stimuli at a specific voltage input and recorded output voltage using the pre-amp microphone. Then, we determined the voltage input needed to generate the desired dB SPL output using the 1 kHz 94 dB SPL speaker as the reference voltage.

### Noise exposure

Unanesthetized and unrestrained mice at P63–70 (for imaging experiments) and at P49–56 (for electrophysiology experiments) were placed within a 5 × 4-in acoustically transparent box and bilaterally exposed to an octave band (8–16 kHz) noise at 100 dB SPL for 2 h noise from a calibrated speaker. For sham-exposed mice, unanesthetized and unrestrained mice were placed within the same box for 2 h, but the noise was not presented. All sham- and noise-exposure experiments were performed at the same time, between 4 and 7 pm.

C57/B6 mice exhibit age-related late-onset hearing loss. However, we conducted our experiments in 8-12 weeks-old mice, and C57/B6 mice do not show age-related hearing loss at this age group[102–106]. Importantly, we used aged-matched sham-exposed mice as controls for all time points and manipulations. Moreover, sham-exposed mice that did not show any changes in either ABR (Fig. 1e, h–j) or cortical response thresholds between P60 and P80 (Figs. 2k, 4k, 5k, and 8k and Supplementary Figs. 2b, 3a, 4a, and 6a), further supporting the absence of age-related hearing loss in our experiments. Taken together, although both treatment groups contain a mutation that could potentially affect vulnerability to noise exposure at some later time point, the different results we observed between sham- and noise-exposed mice, which form the basis of our conclusions, are solely due to the noise exposure, as all other factors, such as age and genetics are equal.

### Auditory brainstem responses

ABR thresholds and ABR wave 1 amplitude were measured with subdermal electrodes in mice at P62-84 under isoflurane anesthesia at a stable temperature (-37 °C) using the RZ6 processor (Tucker–Davis Technologies) as described previously[107]. We recorded ABRs after presenting broadband clicks (1 ms duration, 0–80 dB SPL in 10 dB steps) at a rate of 18.56 per second with a calibrated MF1 speaker (Tucker-Davis Technologies) via a probe tube inserted in the ear canal. We presented each sound 512 times and analyzed the average evoked potential after bandpass filtering the waveform between 300 and 3000 Hz. ABR threshold was defined as the lowest sound intensity that generated ABR wave I amplitudes that were 3 SDs above the baseline noise level. Baseline noise levels were measured using the ABRs obtained at 0 dB SPL sound intensity. ABR wave 1 amplitude was measured from peak to trough levels.

### Distortion product otoacoustic emissions

Mice at P62-84 were anesthetized using isoflurane (3% Induction/1.5% Maintenance, in oxygen) and kept at a stable temperature using a heating pad (-37 °C). Measurements for DPOAE thresholds were taken with the RZ6 processor and BioSigRX software (Tucker–Davis Technologies) as described previously[107]. Tone pairs were presented with an f1 and f2 primary ratio of 1.2 at center frequencies. The f1 and f2 primaries were presented using 2 separate MF1 speakers (Tucker–Davis Technologies) that each presented a frequency into the outer ear canal by using tubing that came together within an acoustic probe to limit artificial distortion. The presentation of these tones into the cochlea results in a distortion product, which is generated by the OHCs and recorded by a sensitive microphone. Recordings were taken at 8, 12,16, 20, and 24 kHz in ascending order from 0 to 80 dB SPL. Each test frequency and intensity were averaged over one hundred sweeps. DPOAE threshold was determined as the lowest intensity that was able to generate a distortion product (2f1–f2) with an amplitude that was at least three standard deviations above the noise floor.

### Adeno-associated virus injections for in vivo imaging

Male or female PV-Cre mice, SOM-Cre mice, and VIP-Cre mice (The Jackson Laboratory) at P49–56 were injected with

AAV9.CaMKII.GCaMP6f. WPRE.SV40 and AAV9.CAG.-Flex.GCaMP6f.WPRE.SV40 into the right auditory cortex as described previously[59,65,108]. Mice were anesthetized with isoflurane (induction: 3% in oxygen, maintenance: 1.5% in oxygen) and secured in a stereotaxic frame (Kopf, Tujunga, CA). Mice's body temperature was maintained at ~37 °C with a heating pad, and the eyes were protected with ophthalmic ointment. Lidocaine (1%) was injected under the scalp, and an incision was made into the skin at the midline to expose the skull. Next, a craniotomy (~0.4 mm diameter) was made over the temporal cortex (~4 mm lateral to lambda). With a micromanipulator (Kopf), a glass micropipette containing AAVs was inserted into the cortex 0.5–0.7 mm past the surface of the dura, and ~500 nL of each viral vector was injected over 5 min. Next, the scalp of the mouse was closed with cyanoacrylate adhesive. Mice were given carprofen 5 mg/kg (Henry Schein Animal Health) to reduce the pain associated with the surgery and monitored for signs of postoperative stress and pain. In case of postoperative stress and pain were observed, mice were given hydrogel (ClearH20) along with carprofen until recovery.

### Animal preparation for acute in vivo wide-field imaging

Twelve to 16 days after AAV injections, mice at P60–80 were prepared for in vivo calcium imaging[59,65,108]. Mice were anesthetized with inhaled isoflurane (induction, 3% in oxygen; maintenance, 1.5% in oxygen) and positioned into a custom-made head holder. Core body temperature was maintained at ~37 °C with a heating pad, and eyes were protected with ophthalmic ointment. Lidocaine (1%) was injected under the scalp, and an incision (~1.5 cm long) was made into the skin over the right temporal cortex. The head of the mouse was rotated ~45° in the coronal plane to align the pial surface of the right temporal cortex with the imaging plane of the upright microscope optics. The skull of the mouse was secured to the head holder using dental acrylic (Lang) and cyanoacrylate adhesive. A tube (the barrel of a 25 ml syringe or an SM1 tube from Thorlabs) was placed around the animal's body to reduce movement. A dental acrylic reservoir was created to hold warm (37 °C) ACSF over the exposed skull. The ACSF contained (in mM)130 NaCl, 3 KCl, 2.4 CaCl$_2$, 1.3 MgCl$_2$, 20 NaHCO$_3$, 3 HEPES, and 10 D-glucose, pH 7.25–7.35, ~300 mOsm. For better optical access to the auditory cortex, we injected lidocaine–epinephrine (2% lidocaine,1:100,000 w/v epinephrine) into the temporal muscle and retracted a small portion of the muscle from the skull. Mice were then positioned under the microscope objective in a sound- and light-attenuation chamber containing the microscope and a calibrated speaker (ES1, Tucker–Davis).

### In vivo wide-field imaging

We performed transcranial imaging to locate the primary auditory cortex (A1) and image sound-evoked activity from specific populations of A1 neurons in awake mice at P60–80. We removed the isoflurane from the oxygen flowing to the animal and began imaging sound-evoked responses after 60 min of recovery from isoflurane[59,65,108]. Sounds were delivered from a free-field speaker 10 cm from the left ear of the animal (ES1 speaker, ED1 driver, Tucker–Davis Technologies), controlled by a digital-to-analog converter with an output rate of 250 kHz (USB-6229, National Instruments). We used ephus[109] to generate sound waveforms and synchronize the sound delivery and image acquisition hardware. We presented 6 or 32 kHz, 50 dB SPL tones to the animal while illuminating the skull with a blue LED (nominal wavelength, 490 nm; M490L2, Thorlabs). We imaged the change in green GCaMP6f emission with epifluorescence optics (eGFP filter set, U-N41017, Olympus) and a 4× objective (Olympus) using a cooled CCD camera (Rolera, Q-Imaging). Images were acquired at a resolution of 174 × 130 pixels (using 4× spatial binning, each pixel covered an area of 171.1 μm$^2$ of the image) at a frame rate of 20 Hz to locate A1 in each animal (see below, Analysis). To locate the A1, we presented low-frequency tones (5 or 6 kHz, 40–60 dB SPL) and imaged the sound-evoked changes in transcranial GCaMP6s fluorescence. Due to the

mirror-like reversal of tonotopic gradients between A1 and the anterior auditory field (AAF)[64,110], these sounds activated two discrete regions of the auditory cortex corresponding to the low-frequency regions of A1 and the AAF (supplementary Fig. 2a). To extract change sound-evoked change in fluorescence ($\Delta F/F$), we normalized the sound-evoked change in fluorescence after the sound presentation ($\Delta F$) to the baseline fluorescence ($F$), where $F$ is the average fluorescence of 1 s preceding the sound onset (for each pixel in the movie). We applied a two-dimensional, low-pass Butterworth filter to each frame of the $\Delta F/F$ movie and created an image consisting of a temporal average of 10 consecutive frames (0.5 s) beginning at the end of the sound stimulus. After localizing A1, we presented broadband sounds (6–64 kHz, 100 ms long) at 30–80 db SPL in 5 db SPL steps from a calibrated speaker (ES1, TDT) and imaged the sound-evoked changes in transcranial GCaMP6f fluorescence signals ($\Delta F/F\%$). Each sound was presented 8–10 times in pseudo-random order.

### In vivo wide-field imaging analysis

A region of interest (ROI, 150–200 mm × 150–200 mm) over A1 was then used to quantify the sound-evoked responses to broadband sounds (6–64 kHz, 100 ms long) sounds. We averaged the fluorescent intensity from all pixels in the ROI for each frame and normalized the $\Delta F$ to the $F$ of the ROI to yield $\Delta F/F$ responses. $\Delta F/F$ responses from 8 to 10 presentations of the same sound level were averaged. Response amplitude was the peak (50 msec window) of the transcranial response that occurred within one second of the sound onset. Response threshold was defined as a sound level that elicits a significant increase in fluorescent signals (two standard deviations above baseline fluorescence $F$). The response gain was defined as the slope of response amplitudes against the sound levels and calculated as the average change in the fluorescence signals ($\Delta F/F\%$) per 5 dB SPL step starting from response threshold[10,11].

### Longitudinal in vivo 2P imaging

After AAV injections into the right AC as described above, we implanted a 3 mm wide cranial glass window over the AC in mice at P49–56 following a published protocol[83,111]. The temporal muscle was separated from the skull sufficiently to allow access to the underlying primary auditory cortex (~4–4.5 mm from the midline). The skull was thinned by careful drilling, and a small area of the skull (~3 mm × 3 mm) was removed, with a 3 mm biopsy punch, to expose the brain. The dura was left intact. Next, a chronic imaging glass window, a 5 mm diameter coverslip, was implanted. The edges between the glass and the skull were sealed with a silicone elastomer (Kwik-Sil). On top of kwik-sil, the edges of the glass and the skull were sealed with dental cement, and a metal head-plate was also affixed to the mice's heads with dental cement to hold them under the 2P-microscope. Mice were head-fixed under a 2P microscope with the head-plate and allowed to acclimate to the rig set up for 30–40 min while we passively played broadband and pure-tone sounds in the background. The next day, after locating A1 using wide-field imaging as described above, we performed 2P imaging of A1 L2/3 neurons (175–225 μm below the pial surface) in awake mice. Mode-locked infrared laser light (940 nm, intensity at the back focal plane of the objective, MaiTai HP, Newport, Santa Clara, CA) was delivered through a galvanometer-based scanning 2P microscope (Scientifica, Uckfield, UK) controlled with scanimage 3.8, using a 40×, 0.8 NA objective (Olympus) with motorized stage and focus controls. We imaged green and red fluorescence simultaneously with two photomultiplier tubes behind red and green emission filters (FF01-593/40, FF03-525/50, Semrock) using a dichroic splitter (DiO2-R561, Semrock) at a frame rate of 5 Hz over an area of 145 × 145 μm and at a resolution of 256 × 256 pixels. We imaged PNs, PVs, SOMs, and VIPs in L2/3 at a depth of ~200 μm from pia. Next, we presented trains of broadband sounds at interstimuli intervals of 5 s (6–64 kHz, 100 ms long) at 30–80 dB SPL in 5 dB SPL steps in pseudo-random order and imaged

the sound-evoked changes in GCaMP6f fluorescence signals ($\Delta F/F\%$). The whole 2P imaging session lasted 20–30 min long, and upon completion, the mice were returned to their cage. To use each neuron as its own control, we manually tracked the same neurons (using blood vessel architecture, depth from the pia, and positions of the neighboring neurons) for 10 days after noise- or sham-exposure and imaged sound-evoked changes in GCaMP6f fluorescence signals to sound-trains. Mice were habituated under the 2P objective for 20-30 minutes a day before the pre-exposure recording sessions. Pre-exposure sessions lasted 2 days, and average responses of individual neurons from both days were used as pre-exposure responses.

## Two-photon analysis

Images were analyzed post hoc using a custom program, and open-source routines, written using Python 3.5 and MATLAB 2007b, 2011b, 2013a, and 2018a as described previously[57]. Before extracting $\Delta F/F$, we used the NoRMCorre software to correct motion artifacts from individual tiff movies[112]. Next, using FISSA, a neuropil decontamination toolbox for calcium imaging signals[113], we selected ROIs around the soma of each L2/3 neuron from the temporal average of all tiff movies from a single recording session. Fluorescence values were extracted from each ROI for each frame, and the mean for each cell was computed. FISSA gave us two vectors of fluorescence values for the somatic and the neuropil. We weighted the neuropil vector by 0.8, as described previously[57,114,115]. The weighted neuropil vector was subtracted from the somatic vector to produce a corrected vector of fluorescence values. This FISSA-corrected fluorescence ($F$) values from each sound trial were then converted to $\Delta F/F$ values by using baseline fluorescence measured 1 s before each sound onset. We then averaged the $\Delta F/F$ values from each sound trail (5–8 trials) to get the mean $\Delta F/F$ from each neuron. To identify the sound-responsive neurons, we used a tone sensitivity index, d-prime ($d'$), from preexposure sessions as described previously[57,58,110]. Briefly, we presented trains of pure tones at interstimuli intervals of 3 s in pseudo-random order that spanned in the range of 4–40 kHz frequencies (500-ms long) in 0.20-octave increments at 30–80 dB SPL in 10 dB SPL steps. For each neuron, we calculated the average response amplitude from responses at and immediately adjacent to the frequency/level combination eliciting the maximum response (preferred frequency, the average of 5 values if the maximum response is observed at dB < 80, 4 values if the maximum response is observed at 80 dB). We then averaged the same number of values selected at random frequency/level locations of the frequency response area (FRA). We took the difference between these averages and iterated this process 1000 times. The tone sensitivity index, d-prime ($d'$), was calculated as the average of the iterated differences, and the neurons with $d' \geq 0$ only were analyzed further. We then used these sound-responsive cells to assess sound-evoked activities, such as response threshold, amplitude, and gain. The sound-evoked responses were measured for 1 s after the sound onset and were defined as significant responses if the sound-evoked changes in $\Delta F/F$ were larger than the mean + 2 standard deviations (SDs) of the baseline fluorescence measured before the sound onset. Peak fluorescence signals during the 1-s period after the sound presentation were quantified as the sound-evoked response amplitude. The response gain was defined as the slope of response amplitudes against the sound levels and calculated as the average change in the fluorescence signals ($\Delta F/F\%$) per 5 dB SPL step starting from response threshold[10,11,20]. Best frequency (BF) was defined as the sound frequency with the maximum average sound-evoked response across all sound levels (30–80 dB SPL)[59].

## Brain slice ex vivo electrophysiology

We recorded intrinsic properties of AC SOMs and VIPs from mice at P50–66 as described previously[64,65]. Due to the lack of cytoarchitectural features, it is challenging to locate the AC in brain slices. Therefore, to localize the AC, we labeled AC corticocollicular (CCol)

L5B PNs (red) projecting to the inferior colliculus by injecting fluorescent retrograde microspheres into the inferior colliculus of SOM-GFP (GIN) or VIP-Cre x Ai14 mice. Briefly, P28–35 male or female mice were injected with red fluorescent retrograde microspheres into the ipsilateral inferior colliculus (IC) (1 mm posterior to lambda and 1 mm lateral, injection depth 0.75 mm). A volume of ~0.12 μL of microspheres was pressure-injected (25 psi, 10–15 ms duration) from capillary pipettes (Drummond Scientific) with a Picospritzer (Parker–Hannifin). The localization of CCol PNs in the AC (Fig. 6b), along with anatomical landmarks, such as the rhinal fissure and the underlying hippocampal formation, allowed us to locate the AC as described previously[63–65]. On the day of recordings, brains were rapidly removed, and coronal slices (300 μm) containing the right AC were prepared in a cutting solution at 1 °C using a Vibratome (VT1200 S; Leica). The cutting solution, pH 7.4, ~300 mOsm, contained the following (in mM): 2.5 KCl, 1.25 NaH$_2$PO$_4$, 25 NaHCO$_3$, 0.5 CaCl$_2$, 7 MgCl$_2$, 7 glucose, 205 sucrose, 1.3 ascorbic acids, and 3 sodium pyruvate (bubbled with 95% O$_2$/5% CO$_2$). The slices were immediately transferred and incubated at 34 °C in a holding chamber for 40 min before recording. The holding chamber contained artificial cerebrospinal fluid (ACSF), pH 7.4, ~300 mOsm containing the following (in mM): 125 NaCl, 2.5 KCl, 26.25 NaHCO$_3$, 2 CaCl$_2$, 1 MgCl$_2$, 10 glucose, 1.3 ascorbic acids, and 3 sodium pyruvate, pH 7.4, ~300 mOsm (bubbled with 95% O$_2$/5% CO$_2$). Next, whole-cell recordings in voltage- and current-clamp modes were performed on slices bathed in carbonated ACSF, which was identical to the incubating solution. For electrophysiological recordings, we used a MultiClamp- 700B amplifier equipped with a Digidata-1440A A/D converter and Clampex (Molecular Devices). Data were sampled at 10 kHz and filtered at 4 kHz. To study the intrinsic properties of SOM neurons, we added the following drugs: 20 μM DNQX (AMPA receptor antagonist), 50 μM APV (NMDA receptor antagonist), and 20 μM SR 95531 Hydrobromide (Gabazine, a GABA$_A$ receptor antagonist). Pipette capacitance was compensated, and series resistance for recordings was lower than 15 M$\Omega$. Series resistance ($R_{series}$) was determined by giving a −5-mV voltage step for 50 ms in voltage–clamp mode (command potential set either at −70 mV or at 0 mV) and was monitored throughout the experiments. $R_{series}$ was calculated by dividing the −5 mV voltage step by the peak current value generated immediately after the step in the command potential. Recordings were excluded from further analysis if the series resistance changed by more than 15% throughout the experiment. Input resistance ($R_{input}$) was calculated by giving a −5-mV step in voltage–clamp mode (command potential set either at −70 mV or at 0 mV), which resulted in transient current responses. The difference between baseline and steady-state hyperpolarized current ($\Delta I$) was used to calculate $R_{input}$ using the following formula: $R_{input} = (−5 \text{ mV}/\Delta I) − R_{series}$. The average resting membrane potential ($V_m$) was calculated by holding the neuron in voltage-follower mode (current clamp, at $I = 0$) immediately after breaking in and averaging the membrane potential over the next 20 s. In the current clamp, depolarizing current pulses (0–450 pA in 50 pA increments of 1-s duration) were used to examine each neuron's basic suprathreshold electrophysiological properties (baseline $V$m was maintained at −70 mV). Action potential (AP) width was calculated as the full width at the half-maximum amplitude of the first resulting AP at rheobase. The AP threshold was measured in the phase plane as the membrane potential at which the depolarization slope exhibited the first abrupt change ($\Delta$slope > 10 V/s). The adaptation ratio was calculated by dividing the instantaneous frequency between the ninth and tenth AP by the instantaneous frequency between the second and third AP (f9/f2).

## Cochlear immunohistochemistry

Mice were deeply anesthetized with isoflurane and sacrificed by decapitation at P77–84. Cochleas were extracted and perfused intralabrynthly with 4% paraformaldehyde in 0.1 M phosphate

buffer as described previously[107]. Cochleas were post-fixed for 2 h at room temperature and decalcified in 120 mM EDTA for 2–3 days at room temperature on a rocker. Decalcified cochleas were then microdissected under a stereomicroscope. Cochlear sections were blocked in 5% normal goat serum with 0.3% Triton X-100 in phosphate-buffered saline (PBS) for 1 h at room temperature. Sections were then incubated in primary antibodies diluted in blocking buffer overnight (18–24 h) at room temperature. Primary antibodies used were anti-myosin VIIa (rabbit anti-MyoVIIa; Proteus Biosciences; 1:500), anti-C-terminal binding protein 2 (mouse anti-CtBP2 IgG1; BD Biosciences; 1:200), and anti-glutamate receptor 2 (mouse anti-GluR2 IgG2a; Millipore; 1:2000). Sections were then washed with PBS and incubated in Alexa Fluor-conjugated fluorescent secondary antibodies (Invitrogen; 1:500) for 2 h at room temperature. The secondaries used, and the dilutions were used were the following: Goat anti-rabbit Pacific Blue (Invitrogen P-10994; 1:500), Goat anti-mouse IgG2a (y2a) (Invitrogen A21131; 1:500), and Goat anti-mouse IgGa (y1) (Invitrogen A21124; 1:500). Sections were again washed in PBS and finally mounted on microscope slides using Prolong Diamond Antifade Mountant (Invitrogen). Cochlear sections were imaged in their entirety at low magnification to reconstruct the cochlear frequency map using an ImageJ plugin provided by Eaton Peabody Laboratories. http://www.masseyeandear.org/research/otolaryngology/investigators/laboratories/eaton-peabody-laboratories/epl-histology-resources/imagej-plugin-for-cochlear-frequency-mapping-in-whole-mounts. This preparation allows us to trace the organ of Corti in its entirety from base to apex, and the plugin superimposes the frequency map on the traced sections. Confocal z-stacks (0.25 mm step size) of the 8, 12, 16, 24, and 32 kHz regions from each cochlea were captured using a Nikon NIS-Elements AR v5.20.00 microscope under a 60× oil immersion lens. Images were imported to ImageJ imaging software for quantification, where maximum projections were rendered from the z-stacks. CtBP2 and GluR2 puncta were counted to identify intact ribbon synapses. Synapses were only considered intact if CtBP2 and GluR2 puncta were juxtaposed. Orphan synapses were defined as CtBP2 puncta that lacked GluR2 puncta. Between 14 and 18 IHCs were included for synapse quantification.

## Computation modeling, LIF network

We consider a four ($a =$ PN, PV, SOM, and VIP) population network of leaky integrate-and-fire (LIF) neurons, where the membrane potential ($V_j^a$) of the $j$th neuron in the population $a$ is governed by the equation

$$\tau_m \frac{dV_j^a}{dt} = -\left(V_j^a - E_L^a\right) + I_j^a(t) + I_{ext}^a(t), \qquad (1)$$

where $E_L^a$ is the resting potential, $\tau_m$ is the membrane time constant, and $I_j^a(t)$ and $I_{ext}^a(t)$ are the recurrent and external synaptic currents, respectively. When $V_j^a(t) \geq V_{th}$, its value is reset to $V_r$ and undergoes a refractory period of length $\tau_r$.

The recurrent synaptic currents are modeled with exponentially decaying synapses

$$\tau_s \frac{dI_j^a}{dt} = I_j^a + \tau_m \left[\sum_b \sum_{k=1}^{N_b} w_{jk}^{ab} \sum_n \delta\left(t - t_k^n\right)\right], \qquad (2)$$

where $\tau_s$ is the synaptic time constant, $w_{jk}^{ab}$ is the strength of the connection from neuron $k$ in population $b$ to neuron $j$ in population $a$ and $t_k^n$ are the spike times of neuron $k$. The probability of a connection from population $b$ to $a$ is given by $p_{a,b}$, and if a connection exists, $w_{jk}^{ab}$ is set to either $w$ or $-gw$ for incoming excitatory or inhibitory inputs, respectively, and 0 otherwise.

$I_{ext}^a(t)$ models the synaptic current from $N_{ext}^a$ Poisson sources with connection strength $w$ and firing rate $r_{ext} = r_{bg}^a + r_{stim}^a$, where $r_{bg}^a$ is the fixed background firing rate and $r_{stim}^a$ is the stimulus firing rate, which depends on the magnitude of the input stimulus (none, low, medium, or high). Instead of explicitly modeling the spiking behavior of this source, we make use of a diffusion approximation

$$\tau_s \frac{dI_{ext}^a}{dt} = -\left(I_{ext}^a - \mu_{ext}\right) + \sigma_{ext}\sqrt{\tau_m}\xi(t), \qquad (3)$$

with $\mu_{ext}^a = wN_{ext}^a \tau_m r_{ext}^a$, and

$$\sigma_{ext}^a = \sqrt{w^2 N_{ext}^a \tau_m r_{ext}^a + \sigma_{inh}^2},$$

where $\sigma_{inh}^2$ is a fixed level of background noise that accounts for additional variability from inhibitory inputs.

## Modeling noise-induced damage and recovery

To model the noise-induced damage seen experimentally, we decrease both the background and stimulus-related firing rates by factors $\gamma, \beta^a < 1$, so that the external firing rate becomes

$$\hat{r}_{ext}^a = \gamma r_{bg}^a + \beta^a r_{stim}^a.$$

Recovery was modeled as a depolarizing or hyperpolarizing current that adjusted the resting potential directly,

$$\hat{E}_L^a = E_L^a + I_{recov}^a.$$

All parameter values for the LIF model can be found in Supplementary Tables 3 and 4. The ratio of excitatory to inhibitory neurons (~20%) and the probability of connections across populations were taken to be in line with experimental results[66], while the rest of the computational parameters were adjusted by Bos et al.[60].

## Mean-field and diffusion approximation

Using the results from [3], we make a diffusion approximation for the recurrent inputs to a neuron. This approximation assumes that the input spike trains follow a Poisson distribution, are uncorrelated, and the amplitude of the depolarization due to each input is small ($w_{ij}^{ab} \ll \theta - V_r$).

Let $N_a$ denote the size of population $a$ and $p_{a,b}$ denote the connection probability of a neuron in population $b$ to a neuron in population $a$. The average number of incoming connections is therefore given by

$$C = \begin{bmatrix} p_{PN,PN}N_{PN} & p_{PN,PV}N_{PV} & p_{PN,SOM}N_{SOM} & 0 \\ p_{PV,PN}N_{PN} & p_{PV,PV}N_{PV} & p_{PN,SOM}N_{SOM} & 0 \\ p_{SOM,PN}N_{PN} & 0 & 0 & p_{SOM,VIP}N_{VIP} \\ p_{VIP,PN}N_{PN} & p_{VIP,PV}N_{PV} & p_{VIP,SOM}N_{SOM} & 0 \end{bmatrix}.$$

Letting $W$ be a matrix of connection strengths

$$W = w \cdot \begin{bmatrix} 1 & -g & -g & -g \\ 1 & -g & -g & -g \\ 1 & -g & -g & -g \\ 1 & -g & -g & -g \end{bmatrix},$$

it follows that the average connectivity between populations is described by

$$J = C \odot W,$$

where $\odot$ denotes the Hadamard product.

Denoting the steady-state firing rates as $\vec{r} = [r_{PN}, r_{PV}, r_{SOM}, r_{VIP}]^T$, we define

$$\vec{\mu}_{eff} = \boldsymbol{J}\vec{r}\tau_m + \vec{\mu}_{ext},$$

$$\vec{\sigma}_{eff}^2 = (\boldsymbol{J} \odot \boldsymbol{W})\vec{r}\tau_m + \vec{\sigma}_{ext}^2,$$

and the mean-field neuronal dynamics are described by the following system of stochastic differential equations

$$\tau_m \frac{dV^a}{dt} = -(V^a - E_L) + I^a(t), \tag{4}$$

$$\tau_s \frac{dI^a}{dt} = -I^a + \mu_{eff}^a + \sigma_{eff}^a \sqrt{\tau_m}\xi(t), \tag{5}$$

where $\xi$ is a white noise term, with zero mean and unit variance density. It follows by [3] that up to the first order in $\sqrt{\tau_s/\tau_m}$, the steady state firing rates are given by

$$r_a = \Phi\left(\mu_{eff}^a + I_{recov}^a, \sigma_{eff}^a\right) = \left[\tau_r + \tau_m\sqrt{\pi}\int_{y_r(\mu_{eff}^a + I_{recov}^a, \sigma_{eff}^a)}^{y_{th}(\mu_{eff}^a + I_{recov}^a, \sigma_{eff}^a)}\Psi(s)ds\right]^{-1} \tag{6}$$

where

$$\Psi(s) = e^{s^2}(1 + erf(s)), \text{ and}$$

$$y_{th,r}\left(\mu_{eff}^a + I_{recov}^a, \sigma_{eff}^a\right) = \frac{V_{th,r} - (\mu_{eff}^a + I_{recov}^a)}{\sigma_{eff}^a} + \frac{\alpha}{2}\sqrt{\frac{\tau_s}{\tau_m}}.$$

Here, $\alpha = \sqrt{2}|\zeta(1/2)|$, $\zeta(\cdot)$ is the Riemann zeta function. Further, the population firing rate dynamics can be described by the following Wilson–Cowan equation

$$\tau\frac{d\vec{r}}{dt} = -\vec{r} + \Phi\left(\mu_{eff}^a + I_{recov}^a, \sigma_{eff}^a\right). \tag{7}$$

**Parameter sweep and stability criteria**

For the three-population model, we estimated the firing rate of the excitatory population $(r_{PN})$ at four stimulus levels $(r_{stim})$ using the mean-field theory and took the excitatory gain to be the slope of the line of best fit $(\hat{m})$

$$r_{PN}(r_{stim}) = m^* \cdot r_{stim} + b^*.$$

We then perform an extensive sweep in the $(\beta^{PN}, \beta^{PV}, I_{recov}^{PN}, I_{recov}^{PV}, I_{recov}^{SOM})$ parameter space. More specifically, for each of these parameters, we consider $n$ ($n = 15$ for $\beta^{PN}$ and $\beta^{PV}$ and $n = 10$ for the I recovery parameters) equally spaced points between bounds defined in supplementary Table 3 and formed a 5-dimensional hypercube consisting of 225,000 interior points/parameter sets. We then estimate the firing rate and gain in the same way as the default case for each parameter set.

A parameter set $(\hat{\beta}^{PN}, \hat{\beta}^{PV}, \hat{I}_{recov}^{PN}, \hat{I}_{recov}^{PV}, \hat{I}_{recov}^{SOM})$ with gain $\hat{m}$ was deemed viable if it demonstrated the following traits observed in the experimental data:

1. $\hat{m} \gtrsim m^*$
2. $(\hat{\beta}^{PN}, \hat{\beta}^{PV}, 0, 0, 0)$, meaning the parameter set without recovery show decreased gain from the default value $m^*$
3. The firing rate of all populations monotonically increased with stimulus strength

4. A low PN response threshold, defined to be >1 Hz for the low stimulus value
5. Stable dynamics (see details below)

Some parameter sets were immediately discarded due to the stability criteria because the mean-field theory failed to converge to a stable solution. However, for some of the considered parameter sets, we found examples where the converged mean-field theory disagreed with the corresponding result from the spiking model due to the spiking model being pushed into an unstable, synchronous, and heavily correlated regime. Due to this disagreement, we wanted to discard such parameter sets from the analysis.

As stated previously, the mean-field theory assumes that the spiking dynamics lays in an asynchronous regime, and this disagreement arises when this assumption breaks down. Unfortunately, it is not straightforward to use the theory to predict exactly when this disagreement will occur for a given parameter set[68,116]. Here, we use the eigenvalues of the Jacobian of the deterministic model to provide a conservative and unbiased threshold to disregard such parameter sets. Specifically, we linearize Eq. (1) around the fixed point

$$\tau\frac{d\vec{r}}{dt} = -\vec{r} + \boldsymbol{A}\vec{r},$$

and then disregard parameter sets where $\max\Re(\lambda(\boldsymbol{A} - \boldsymbol{I})) = \kappa > \kappa_{thres}$. The deterministic system is unstable when $\kappa > 0$, but in order to discard parameter sets that lead to the disagreement between the theory and the spiking model described above, we consider the conservative threshold of $\kappa_{thres} = -0.7$.

The methods and criteria are the same for the four-population model, but having eliminated intrinsic changes in the SOM population experimentally and adding the VIP population to the model, we consider the parameter space $(\beta^{PN}, \beta^{PV}, \beta^{VIP}, I_{recov}^{PN}, I_{recov}^{PV}, I_{recov}^{VIP})$. Following the results of the first model, the parameter space hypercube was also adjusted to only consider positive values of $I_{recov}^a$. In total, 15,625 total parameter sets were considered.

**Numerical details**

The spiking network is implemented in Euler's method with a timestep of 0.01 ms. Each trial consisted of 5 s, and the steady state firing rates were computed after averaging the spiking activity of the neurons across each population after discarding the first second of each trial. The firing rates for the mean-field theory were found via fixed point iteration of Eq. (1), which halted when $||\vec{r}^{n+1} - \vec{r}^n|| < 10^{-5}$.

**Statistics and reproducibility**

For statistical comparisons between two independent groups that passed the Shapiro–Wilk normality test, we used unpaired t-tests. Otherwise, we used the Mann–Whitney rank-sum test, Kruskal–Wallis one-way analysis of variance, or Friedman test for non-normally distributed data. For comparisons between multiple groups having within-subject factors, a repeated measures two-way ANOVA test was used, and Bonferroni corrections were used for multiple two-sample post hoc comparisons among sample groups; the significance level ($\alpha = 0.05$) of the test was corrected via scaling by the reciprocal of the number of comparisons. A greenhouse–Geisser correction was used when the assumption of sphericity was violated. A permutation test (Wasserman, 2004) was used for two sample comparisons. Samples for which 5000 of 100,000 random permutations of the data resulted in mean differences greater than the observed difference in sample means were considered significant ($p$, 0.05). Significance levels are denoted as $*$, $p < 0.05$. All the statistical tests were two-sided. The details of statistical tests are described in the figure legends. Group data are presented as mean ± SEM. Sample sizes were not predetermined

**Article**

by statistical methods but were based on those commonly used in the field. The images shown in Figs. 1l; 2i; 4i; 5i; 8i; and Supplementary Fig. 1b are representative images from experiments that were repeated as many times as shown in the corresponding summary graphs.

## Rigor and transparency
Behavioral, in vitro electrophysiology, and histology experiments were conducted and analyzed in a blind mode regarding noise- and sham-exposed conditions. For in vivo imaging experiments, analysis was also done in a blind mode. Although the experimenter was not "blind" during the acquisition of the in vivo imaging experiments, those experiments involved identical and automated signal detection, inclusion, and analysis for both noises- and sham-exposed mice (as described in "Methods"). Thus, the experimenter did not have any influence over the experiment. Thus, between the data acquisition and analyses, all experiments and analyses are transparent, rigorous, and reproducible.

## Reporting summary
Further information on research design is available in the Nature Portfolio Reporting Summary linked to this article.

## Data availability
The experimental processed data (source data) that support the findings of this study are provided in this paper. Upon request, the raw data files will be made available by the corresponding authors. Source data are provided in this paper.

## Code availability
The simulation data, the code that produced the modeling panels (Fig. 3, Fig. 7, and Supplementary Fig. 5), and the code that runs the spiking model and mean-field theory can be found on Zenodo. The code is written in a combination of C and MATLAB. Custom-written MATLAB codes to exact and analyze 2P data can be found on Zenodo.

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

## Acknowledgements

We thank Dr. Patrick Cody for help with data analysis. We thank our funding sources: NIH awards DC019618 and R01 EB033172, (T.T), Hearing Health Foundation award 855358 (MK), Swartz Foundation Fellowship for Theory in Neuroscience, and the Burroughs Wellcome Fund's Career Award at the Scientific Interface (G.H), and grants from the Simons Foundation Collaboration on the Global Brain (B.D).

## Author contributions

M.K. and T.T. designed the study. M.K. performed in vivo experiments and analyzed data. G.H. and B.D. designed and programmed computational modeling. S.K. and Y.Z. performed electrophysiology experiments. L.L.B. and E.W. performed and analyzed ABR and DPOAE experiments. B.B. performed cochlear histology. M.K., G.H., B.D., and T.T. wrote the manuscript.

## Competing interests

The authors declare no competing interests.
