## [Peer Review File · Nature Communications]

REVIEWER COMMENTS

Reviewer #1 (Remarks to the Author):

This study describes the functional properties of auditory cortex pyramidal and inhibitory neurons following noise-induced hearing loss, and reports that the recovery of perceptual thresholds is associated with inhibitory interneuron plasticity. The documented changes following noise-induced hearing loss include VIP interneuron suppression of SOM activity, which should lead to decreased SOM inhibition of both PNs and PV interneurons, and this is interpreted as exerting a stabilizing influence on network activity. A strength of the study is the number of technically sophisticated measures of inhibitory neuron activity in vivo. However, weaknesses in the experimental design limit the conclusions that can be drawn from the findings.

Main comments:

(1)

The experimental approach (cell-type specific manipulations with Cre driver lines) required the use of C57/B6 mice, a strain that displays early onset hearing loss. Therefore, the manipulation may involve one form of hearing loss (noise damage) added to a second form of hearing loss (the 57 genetic deficit). Furthermore, the magnitude of noise-induced hearing loss is greater in young mice (below 16 weeks postnatal), at least in normal hearing strains (Kujawa and Liberman, 2006). The reason for raising this information is that I was not able to find the animal ages, either at exposure or at testing. There were a few exceptions, but these did not help me piece together the full experimental design (e.g., when reporting on brain slice recordings, this statement appears: "P28-35 male or female, SOM-GFP (GIN) mice were injected with red fluorescent retrograde microspheres"; when reporting the behavioral training, this statement appears: "Six to seven-week-old C57B6 mice were initially trained to cross from one side of the shuttle-box to another"). Since age is a meaningful variable in hearing loss studies, it is essential to provide this information. Ideally, this information is placed in the "Animals" section of the Methods. If information about animal age was clearly presented in the text, and I simply missed it, then please accept my apology.

(2)

There is no direct behavioral evidence to support the premise that perception was influenced by, or recovered following, noise exposure. The experimental design compares a behavioral measure of threshold obtained 10 days post-noise exposure with ABR measures of threshold obtained pre-noise exposure and at 1-10 days after exposure. The manuscript appears to assume that the ABR threshold shifts reported between pre-noise and 1 day post-noise serve as an adequate proxy for behavior.

There are a few reasons to question this assumption. First, noise damage with this paradigm (8-16 kHz, 100 dB SPL, 2 hours) produces a non-uniform effect along the cochleotopic axis, causing the largest shifts in threshold and loss of synapses at the most basal region (Kujawa and Liberman, 2009). Since ABR recordings were obtained with clicks (line 585), it is not clear how the click-evoked thresholds correlate with the frequency-specific pattern of peripheral threshold shifts. Furthermore, since the behavioral test used 50 ms noise-bursts (lines 568-9), it is not clear how the perceptual thresholds relate to the click-evoked ABR thresholds, and it is not even clear to what extent the ABR paradigm assesses a noise-damaged regions of the cochlea.

Direct evidence of a behavioral deficit, followed by recovery, would require that animals be tested for perceptual thresholds pre-exposure to establish a normative baseline, at 1 day

post-exposure to establish a deficit, and then at 10 days to establish recovery. Ideally, these tests would use one or more tones so that the data could be compared to the pattern of ribbon synapse loss, and ABRs would be obtained with pure tones to establish the audiometric pattern of threshold elevation.

There is a second, more global problem with the argument that auditory cortex plasticity can explain a recovery of perceptual thresholds. There is no direct evidence that mice use auditory cortex to perform the behavioral test for threshold determination. This would require an inactivation of auditory cortex during a behavioral testing session. If auditory cortex is used for sound detection, then all of the physiological measurements would be relevant. However, there is not enough evidence to accept this assumption.

(3)

Similar to the behavioral and ABR measurements, the imaging experiments should have taken into account the differences in neuron thresholds that would occur as a function of the non-uniform cochlear damage. Fig 1J suggests that these differences would be restricted to frequencies at or above 16 kHz. Without this information it is difficult to interpret the technically elegant measurements. For example, did auditory cortex PNs or inhibitory interneurons with center frequencies associated with an undamaged region of the cochlea display changes of any sort following noise damage and, if so, why? One would expect that auditory cortex neurons would display a diverse range of outcomes that would be correlated with tonotopic position (i.e., damage). This information is would be required to draw compelling conclusions about the network.

(4)

There are two conceptual problems. First, the relationship between the documented changes in auditory cortex and the perceptual phenomenon to be explained (normal behavior thresholds despite abnormal ABR thresholds), or any other perceptual impact of hearing loss, remains enigmatic. The general idea is that reduced SOM activity leads to both increased PV to PN inhibition and reduced SOM to PN inhibition, and this is said to increase network stability. How does network stability explain detection threshold? Would it be possible to provide a plausible measure of network stability from the data that you have collected? Second, the general model is based on the sound-evoked responses recorded from the soma of each neuron population, but the most pivotal parameter is synaptic strength at each connection. It is well beyond the scope of this study to collect this information, but these properties should be considered in more depth.

Minor comments:

(5)

The rationale for providing both the acute widefield and longitudinal data sets of threshold responses from A1 PNs (Figs 2d vs k) is unclear, especially since they are not consistent. For example, there is a 40 dB shift at 1 day for panel d, but only a 10 dB shift at 1 day for panel k. Another example is that there is partial recovery at 10 days for panel d, but complete recovery for panel k. How do you interpret the differences between the two data sets? I would have assumed that the longitudinal data is more robust and interpretable than the widefield data. In fact, the manuscript notes several limitations of the widefield imaging data (lines 167-170) which serve as a rationale for collecting the superior individual neuron data. It is not clear what is learned from the widefield data that is critical to the questions being addressed. What would be lost if only the L2/3 data were to be presented?

The same concern emerges for Figure 4 (PV neuron widefield vs longitudinal 2-photon

imaging; panels d vs k), Figure 5 (SOM neuron widefield vs longitudinal 2-photon imaging; panels d vs k), and Figure 8 (VIP neuron widefield vs longitudinal 2-photon imaging; panels d vs k).

(6)

The introduction gives the impression that recovery of synaptic function is a common occurrence following hearing loss, so it is worth clarifying the types of hearing loss that are being considered when making global statements. Both developmental hearing loss and presbycusis are each associated with many permanent changes, including to excitatory and inhibitory synapses.

Reviewer #3 (Remarks to the Author):

This manuscript by Kumar et al. examined how different neuronal cell types in auditory cortex respond to peripheral damage (noise-induced hearing loss, NIHL), using modeling and Ca²⁺ imaging techniques. The main findings are: 1) pyramidal (PN) and parvalbumin (PV) neurons recover their threshold and response amplitude 10 days after NIHL with an increased gain; 2) SOM neurons are suppressed without recovery, contributing to the increased gain of PN and PV neurons; 3) the suppression of SOM neurons could be due to a strong recovery of VIP neurons. This is probably the first study to carry out such extensive investigations of roles of different cell types in cortical rehabilitation, thus is a valuable addition to the scientific literature. Experiments are in general well executed. The reviewer has some specific points to improve the manuscript.

Major:

1) The computation models the authors used are useful and generate guidance to look into specific features of cortical cell types with experimental approaches. In the four-population model, the VIP population is designed to receive feedforward input just like PN and PV cells. However, the Ji et al. (Cerebral Cortex 2016) study shows that VIP neurons (even in layer 2/3) do not receive direct thalamic input. Will removing this feedforward input affect the result?

2) This manuscript implies that the recovery of VIP neuron responses is the main driving force for the increased gain of PN neurons via suppressing SOM neurons. However, it fell short of explaining what factors drive the recovery of VIP neurons. The authors can do similar ex vivo recording experiments in VIP neurons after NIHL, at least to provide some possible mechanisms.

Minor:

Please provide subtitles in the Result section.

In Fig.2f, please label with marks to indicate whether there is a significant reduction of AN gain after NIHL.

Fig. S2j, Fig. S4g, please label the X-axis.

Fig.3f, what is the unit for the X-axis?

How old are these animals? Please indicate in the Methods.

In the figure legends, brackets are not consistent, for example sometimes "a)" is used, and some others "(a)" is used. In the line 1266, where is the figure (p)? It should be figure (i).

Line 47, "...due to increased cortical gain, the sensitivity of neuronal responses against sound levels.". Add "which is" before "the sensitivity...".

Line 161, "...remained increased elevated...". Remove "elevated".

Line 444, "cochlear damage may contribute to the hearing problems peripheral damage...". Add "after" before "peripheral damage".

Reviewer #4 (Remarks to the Author):

In this manuscript, Kumar et al. explore the plastic changes of cortical neuronal subtypes following noise-induced hearing loss (NIHL). They start by confirming, both with behavioral experiments, ABR and DPOAE measurements, immunohistochemistry, wide field transcranial, and 2P imaging, that 2 hours 100 dB SPL noise exposure induces hearing loss that is followed by plastic changes in the principal neurons (PN) of auditory cortex. They then ask what the core underlying mechanism of this PN plasticity is by exploring the effect of NIHL on 3 types of interneurons (PV, SOM, and VIP neurons). Using wide field and 2P imaging in mice expressing GCAMP6f in the 3 populations of interest, as well as modeling and in vitro patch-clamp recordings, they show that PV and VIP neurons, like PN neurons, recover sound-evoked activity 10 days after noise exposure, whereas SOM neurons decrease their activity. Together, these results show cell-type specific plasticity of cortical neurons following peripheral lesion.

Although cortical recovery following peripheral damage has already been described, the underlying mechanisms are not known, and this study adds relevant, new, and important information to the topic. The experimental approaches are diverse and carefully conducted, and the results are solid and convincing. The study is also very well explained and follows logical steps of reasoning. The manuscript is well written. The only weak point of the manuscript is the interpretation of the results. As such, they are "just" suggestions, but a few additional experiments could significantly strengthen them. Please find my suggestions below:

Major concerns

- 1) The possible interpretation, as suggested for example on l.29-30 of the abstract, would be much more convincing if data could prove this. This circuit relies on the canonical circuit illustrated in Figure 7a – but the reality in an auditory cortical column is much subtler than this, and strongly dependent on context (see the review by Studer et al., Neuro Bio Rev 2021 for some examples). If manipulating VIP neurons (for example by inhibiting them with optogenetics) would lead to activity similar to pre-NE data, I would be much more convinced that this is indeed a possible mechanism. If such an experiment is not possible, I would advise to keep a very interpretative version of this mechanism (the formulation on l.299 (...support the notion that...) is preferred over the one of l.303 (...contributes to..)).**
- 2) The in vitro data on SOM neurons is very nice and convincing, but I wonder why putative changes in intrinsic properties have only been explored in SOM neurons, and not in the 3 other cell types? I would have been particularly interested in knowing whether VIP neurons change their intrinsic properties upon NIHL. If they do not change, could the authors speculate about the origin of the increased activity in VIP neurons (that could then explain the changes in activity in the 3 other cell types)? This would complete the suggested mechanism, for example discussed on l.488-490.**
- 3) Given that it is widely known that the inhibitory neurons include VIP, SOM, and PV neurons (as correctly cited on l.60-62), I wonder why the model did not include VIP neurons from the beginning. In other words, what is the reasoning for having a model without VIP neurons (Figure 3), instead of starting directly with a model including this interneuron subtype (Figure 5)?**

Minor concerns

- 1) **Figures 4 and 5: Although similar, the wide-field imaging (Fig 4e and 5e) does not give the same results as the 2P responses (Fig 4m and 5m). Could the authors speculate why this would be the case?**
- 2) **L.26-27. Why not put PN, PV, and VIP together in these 2 lines?**
- 3) **L.47. What is sensitivity of neuronal responses “against sound levels”?**
- 4) **L.62. It would be appropriate to also cite a recent review on inhibitory neurons particularly in the auditory system: Studer & Barkat, Neuroscience and Biobehavioral Reviews 2021**
- 5) **The sentence in l.91-95 would be easier to read if it would be shorter (or separated in several sentences).**
- 6) **L.140 Fig. 2a, b instead of Fig. 2ab**
- 7) **L.159. Although it might be in the methods, it would be nice to here briefly mention how the response gain of sound-evoked activity is quantified.**
- 8) **L. 179. Similarly, it would be nice to briefly explain how tuning strength is quantified.**
- 9) **L.230. What is an “extensive, brute force parameter sweep”?**

Response to Reviewers

We thank the reviewers for the generally positive comments. We also appreciated the raised concerns, which helped us to improve the paper. We performed substantial revisions and addressed all the concerns. Namely, we performed many additional experiments and analyses to address experimentally the reviewers' concerns (**new Rebuttal Fig. 1; new Figures: 1h-j; supplement Fig. 1a; Fig. 2q and r; Fig. 4q and r; Fig. 5q and r; Supplement Fig. 5; Fig. 8q and r; and Figure 9**). Of course, we also revised the manuscript accordingly (revised parts have colored fonts). Please see our point-by-point response for details.

Reviewer #1

We thank Reviewer 1 for the constructive and helpful comments. Reviewer 1 evaluated the “strength of the study is the number of technically sophisticated measures of inhibitory neuron activity in vivo. However, weaknesses in the experimental design limit the conclusions that can be drawn from the findings”. To address these concerns, we performed additional experiments and analyses to strengthen our experimental design and conclusions (**new Rebuttal Fig. 1, see below; new Figures: Fig. 2q and r, Fig. 4q and r, Fig. 5q and r, Fig. 8q and r**). Moreover, we revised the manuscript accordingly. Below we cite the raised concerns and provide our point-by-point responses.

Main Comments

Comment 1: The experimental approach (cell-type specific manipulations with Cre driver lines) required the use of C57/B6 mice, a strain that displays early onset hearing loss. Therefore, the manipulation may involve one form of hearing loss (noise damage) added to a second form of hearing loss (the 57 genetic deficit). Furthermore, the magnitude of noise-induced hearing loss is greater in young mice (below 16 weeks postnatal), at least in normal hearing strains (Kujawa and Liberman, 2006). The reason for raising this information is that I was not able to find the animal ages, either at exposure or at testing. There were a few exceptions, but these did not help me piece together the full experimental design (e.g., when reporting on brain slice recordings, this statement appears: “P28-35 male or female, SOM-GFP (GIN) mice were injected with red fluorescent retrograde microspheres”; when reporting the behavioral training, this statement appears: “Six to seven-week-old C57B6 mice were initially trained to cross from one side of the shuttle-box to another”). Since age is a meaningful variable in hearing loss studies, it is essential to provide this information. Ideally, this information is placed in the “Animals” section of the Methods. If information about animal age was clearly presented in the text, and I simply missed it, then please accept my apology.

Response: We updated the methods section to specify the animal age more clearly. For experiments shown in **Fig. 1** describing the animal model of noise-induced peripheral damage, male and female PV-Cre, SOM-Cre mice, and VIP-Cre mice with C57/B6 mice backgrounds (The

Jackson Laboratory) were noise-exposed at P63-70. ABR, DPOAE, and histology were performed at P62-84. For wide-field and 2P experiments shown in **Figs. 2, 4, 5 and 8** male and female PV-Cre, SOM-Cre, and VIP-Cre mice were injected with AAVs at P49-56, noise-exposed at P63-70, and imaging was performed at P60-80. For in vitro electrophysiological experiments assessing intrinsic properties in SOM neurons (SOMs) shown in **Fig. 6**, male and female SOM-GFP (GIN) mice with C57/B6 mice backgrounds were injected with retrograde beads at P28-35, noise-exposed at P49-56, and electrophysiology was performed at P50-66. For in vitro electrophysiological experiments assessing intrinsic properties in VIP neurons (VIPs) shown in **Fig. 9**, male and female offspring from VIP-Cre x Ai14 (The Jackson Laboratory) were injected with retrograde beads at P28-35, noise-exposed at P49-56, and electrophysiology was performed at P50-66.

We agree with the reviewer that C57/B6 background exhibits age-related late-onset hearing loss. However, we conducted our experiments in 8-12 weeks-old mice, and at that age group C57/B6 mice do not show an age-related hearing loss¹⁻⁵. Moreover, in our study, we used sham-exposed mice that did not show any changes in either ABR (**Fig. 1e, h-j**) or cortical response thresholds (**Fig. 2k, 4k, 5k, and 8k and supplementary fig. 2b, 3a, 4a, and 6a**), further supporting the absence of age-related hearing loss in our experiments. Importantly, we used aged-matched sham-exposed mice as controls for all time points and manipulations. Thus, the results we observe between sham- and noise-exposed are solely due to the noise exposure, as all other factors, such as age, genetics, etc., are equal.

Comment 2: There is no direct behavioral evidence to support the premise that perception was influenced by, or recovered following noise exposure. The experimental design compares a behavioral measure of threshold obtained 10 days post-noise exposure with ABR measures of threshold obtained pre-noise exposure and at 1-10 days after exposure. The manuscript appears to assume that the ABR threshold shifts reported between pre-noise and 1-day post-noise serve as an adequate proxy for behavior.

There are a few reasons to question this assumption. First, noise damage with this paradigm (8-16 kHz, 100 dB SPL, 2 hours) produces a non-uniform effect along the cochleotopic axis, causing the largest shifts in threshold and loss of synapses at the most basal region (Kujawa and Liberman, 2009). Since ABR recordings were obtained with clicks (line 585), it is not clear how the click-evoked thresholds correlate with the frequency-specific pattern of peripheral threshold shifts. Furthermore, since the behavioral test used 50 ms noise-bursts (lines 568-9), it is not clear how the perceptual thresholds relate to the click-evoked ABR thresholds, and it is not even clear to what extent the ABR paradigm assesses a noise-damaged regions of the cochlea.

Direct evidence of a behavioral deficit, followed by recovery, would require that animals be tested for perceptual thresholds pre-exposure to establish a normative baseline, at 1-day post-exposure to establish a deficit, and then at 10 days to establish recovery. Ideally, these tests would use one or more tones so that the data could be compared to the pattern of ribbon synapse loss, and ABRs would be obtained with pure tones to establish the audiometric pattern of threshold elevation.

Response: We agree with the reviewer, and we performed the requested experiment.

However, given the next comment from Rev#1 and comments from the other Revs#2 and 3 and our evaluation of the feedback, we opted to remove the behavior (perceptual hearing thresholds) from the manuscript. Here is why. In general, we would like to apologize if our writing conveyed that our results support that auditory cortex plasticity can explain the recovery of the perceptual detection thresholds after noise exposure. As the revised title of our manuscript “Cell-type-specific plasticity of inhibitory interneurons in the rehabilitation of auditory cortex after peripheral damage” indicates, the goal of our study is to identify how the response properties to sound and the intrinsic

[redacted]

excitability of different cortical interneuronal subtypes change during the recovery of cortical sound-processing. Indeed, we found cell-type-specific cellular excitability and response thresholds and gain of A1 neurons after noise-trauma. This distinct, cell-type-specific plasticity supports the notion that PVs, SOMs, and VIPs play distinct roles in cortical recovery after peripheral damage. Namely, after noise trauma, we found enhanced sound-evoked activity in PNs (**Fig. 2**), PVs (**Fig. 4**), and VIPs (**Fig. 8**), but reduced sound-evoked activity in SOMs (**Fig. 5**). Based on the known sequentially organized inhibitory cortical network, where VIPs neurons inhibit SOMs, SOMs inhibit PVs and PNs, and PVs inhibit other PVs and PNs, our results are consistent with the notion that the underlying $SOM \rightarrow PV \rightarrow PN$ and $SOM \rightarrow PN$ circuits support a cell-type-specific plasticity mechanism in which, robust PV activity provides network stability by balancing PN activity; and vastly decreased SOM activity allows for increased PV and PN gain, which supports stability and high gain. The $VIP \rightarrow SOM \rightarrow PN$ disinhibitory pathway completes the task, whereby robust VIP activity enables reduced SOM activity. Although the causality or even the contribution between the proposed roles of PVs, SOMs, and VIPs and recovered A1 sound processing were not tested here, it will be evaluated in future work. Moreover, we are aware that the recovery of cortical sound processing after noise trauma is likely also influenced by subcortical plasticity mechanisms, which are also outside the scope of this study. In sum, our study is the first to identify the cell-type-specific plasticity of the different cortical IN subtypes in cortical recovery after noise trauma; and it suggests distinct roles for these interneurons in this process. As such, these results highlight a novel strategic, cooperative, and cell type-specific plasticity program that may contribute to restoring cortical sound processing after cochlear damage, and thus, may provide novel cellular targets that may also aid in the development of pharmacotherapeutic or rehabilitative treatment options for impaired hearing after NIHL.

This is all stated in our revised discussion and throughout the manuscript.

Regarding the relevance of these mechanisms on underlying perceptual recovery, our findings, are consistent with the notion that cortical recovery might contribute to the recovery of perceptual detection thresholds after noise trauma. Previous studies are also consistent with this notion^{7,9,10}. However, causality was neither studied nor supported here. In future studies, which are outside the scope of this manuscript, we plan to address the potential causal role of our identified cortical plasticity both for cortical and perceptual recovery. This is why we opted to remove the behavioral data, which will be part of our next study where we will explore the influence of our discovered mechanisms and proposed roles of the different neuronal types on perceptual recovery.

In the revised manuscript, we clearly state the focus, the strengths and limitations, and the interpretation of our study in this context.

Comment 3: There is a second, more global problem with the argument that auditory cortex plasticity can explain a recovery of perceptual thresholds. There is no direct evidence that mice use auditory cortex to perform the behavioral test for threshold determination. This would require an inactivation of auditory cortex during a behavioral testing session. If auditory cortex is used for sound detection, then all of the physiological measurements would be relevant. However, there is not enough evidence to accept this assumption.

Response: Please see previous response.

Comment 4: Similar to the behavioral and ABR measurements, the imaging experiments should have taken into account the differences in neuron thresholds that would occur as a function of the non-uniform cochlear damage. Fig 1J suggests that these differences would be restricted to frequencies at or above 16 kHz. Without this information it is difficult to interpret the technically elegant measurements. For example, did auditory cortex PNs or inhibitory interneurons with center frequencies associated with an undamaged region of the cochlea display changes of any sort following noise damage and, if so, why? One would expect that auditory cortex neurons would display a diverse range of outcomes that would be correlated with tonotopic position (i.e., damage). This information would be required to draw compelling conclusions about the network.

Response: We agree with the reviewer. To address this concern, we performed additional pure tones ABR and 2P data analysis to correlate the peripheral damage with cortical plasticity.

Although noise exposure caused the synaptopathy at the cochlear region 16 kHz or above (**now Fig. 1m**), we found that noise exposure significantly increased the ABR response thresholds for 8-32 kHz tones, which remained elevated even 10 days after noise exposure (**Fig 1h-j**), suggesting persistent and widespread cochlear damage across the tested frequencies. We also observed frequency-specific damage at 10 days after noise exposure, where we found a larger ABR threshold shift at 32 kHz compared to 8 kHz tones (**Fig. 1 supplement 1a**).

To correlate the cochlear damage with the plasticity in the sound-evoked activities of individual L2/3 neurons, we analyzed the response threshold and gain of PNs as a function of their pre-exposure best frequencies (BF), the sound frequency with the maximal sound-evoked response¹¹ (**Fig. 2q, r**; Methods). Irrespective of their BF, PNs showed increased response thresholds 1 day after NIHL (**Fig. 2q**). However, 10 days after NIHL, PNs with low- (4-8.6 kHz) and mid-BF (10-21.6 kHz) showed complete recovery in their response thresholds, but PNs with high-BF (25.2 – 40 kHz) did not (**Fig. 2q**). Consistent with previous results¹⁰, our results support the notion that that PNs with BFs corresponding to the high-frequency cochlear region, which showed more damage compared to the low-frequency region (**Fig. 1 supplement 1a**), do not recover their response thresholds completely. In terms of gain, irrespective of their BF, PNs showed increased gain 1-10 days after NIHL (**Fig. 2r**) suggesting that PNs with BF across all the tested frequency regions of the cochlea show increased gain after NIHL. Finally, we did not observe a change in either PN threshold (**Fig. 2k**) or gain (**Fig. 2n**) in sham-exposed mice (218 neurons from 5 mice, **Fig. 2k, n and supplement Fig. 2f-j**). In sum, whereas the recovery in response threshold was less robust for PNs with high BF, the increase in gain was more uniform across all frequencies. Together, our results show that despite the persistent peripheral damage, A1 L2/3 PNs show recovered response thresholds and response amplitudes, and increased response gain.

To correlate the peripheral damage to the PV sound-evoked properties, we analyzed the response threshold and gain of PVs as a function of their BF (**Fig. 4q, r**). Irrespective of their BF, PVs showed increased response thresholds 1 day after NIHL recovered to pre-exposure levels by 10 days (**Fig. 4q**). Similarly, we found increased gain in PV neurons with low- to high-BF, 1 day after NIHL that remained increased for 10 days (**Fig. 4r**). These results suggest that PVs with BF across all the frequency regions displayed recovered thresholds and increased gain after NIHL, suggesting that PV recovery is robust and not correlated with the peripheral damage. We did not observe any changes in the response threshold, amplitude, and gain of L2/3 PVs in sham-exposed mice (80 neurons from 7 mice, **Fig. 4k, n and supplement Fig. 3c-g**). These results demonstrate that, in response to peripheral damage, A1 L2/3 PVs match the recovery of PNs. Consequently, this recovery is not consistent with PV contribution to the increased PN gain after

recovery from NIHL. However, this result and our computational modeling results (**Fig. 3e, f**) are consistent with the notion that PVs may act as stabilizers of A1 network activity after noise trauma.

When we analyzed SOM response threshold and gain as a function of their BF (**Fig. 5q, r**), we found that irrespective of their BF, SOMs showed increased response thresholds 1 and 3 days after noise exposure (**Fig. 5q**). However, 10 days after noise exposure, SOMs with mid- and high-BF showed partial recovery in their response thresholds (**Fig. 5q**). These results suggest that SOMs with BF corresponding to the high-frequency region of the cochlea, which showed more damage compared to the low-frequency region (**Fig. 1 supplement 1a**), show partial recovery in their response thresholds after NIHL. Since SOM response threshold plasticity is opposite to PN threshold plasticity [Low-BF: PNs recover but not SOMs – High-BF: SOMs recover but not PNs (**Fig. 2q and 5q**)], these results are consistent with the notion that SOM threshold plasticity is linked with PN threshold plasticity. In terms of gain, consistent with the overall no change in SOM gain (**Fig. 5n**), we did not find a change in the gain of SOMs when binned as per their BF (**Fig. 5r**). Finally, we did not observe a change in SOM response threshold, amplitude, and gain in sham-exposed mice (**Fig. 5k, n and supplement Fig. 4c-g**, 42 neurons from 9 mice). In total, these results are consistent with the second modeling prediction that SOM responses are suppressed during recovery from NIHL. Together, these results and our computational modeling results (**Fig. 3e, f**) are consistent with the notion that the reduced SOM activity disinhibits PVs and PNs, thus allowing for high PV and PN response gain after NIHL.

When we analyzed VIP response threshold and gain as a function of their BF (**Fig. 8q, r**), we found that VIPs with low- and mid-BF showed increased response thresholds 1 day after NIHL, which were recovered to baseline levels by 10 days after NIHL (**Fig. 8q**). However, VIPs with high-BF showed reduced recovery in their response thresholds (**Fig. 8q**). These results suggest that VIPs with BF corresponding to the high-frequency region of the cochlea, which showed more damage compared to the low-frequency region (**Fig. 1 supplement 1a**), do not recover their response thresholds completely. In terms of gain, VIPs showed increased gain after NIHL across all the frequencies (**Fig. 8r**). Taken together, our results support a strong recovery of VIP activity after noise trauma, even surpassing the control activity. Because VIPs inhibit SOMs⁶ (**Fig. 8b**), these results are consistent with the circuit mechanism where robust VIP activity enables SOM suppression, which in turn leads to high PN and PV gain.

All the above sections are now included in the revised results section.

Comment 5: There are two conceptual problems. First, the relationship between the documented changes in the auditory cortex and the perceptual phenomenon to be explained (normal behavior thresholds despite abnormal ABR thresholds), or any other perceptual impact of hearing loss, remains enigmatic. The general idea is that reduced SOM activity leads to both increased PV to PN inhibition and reduced SOM to PN inhibition, and this is said to increase network stability. How does network stability explain detection threshold? Would it be possible to provide a plausible measure of network stability from the data that you have collected?

Response: We thank the reviewer for highlighting these conceptual points of confusion.

To address the first set of questions, note that detection threshold is inherently tied to cortical gain¹², which in turn has previously been tied to network stability¹³. High cortical gain implies that a weak stimulus results in a large network response (compared to spontaneous), and hence such a network would have a low detection threshold. One way to increase cortical gain is to decrease

the amount of inhibition within a network, allowing any excitation to elicit a stronger response. However, a problem arises in an inhibition-stabilized network with a single inhibitory subtype, since removing inhibition also pushes the network closer to instability (i.e. runaway activity).

What we have observed with the PN, PV, SOM circuit is consistent with a division of labor among interneurons to achieve the increased cortical gain without risking the network destabilization¹³. Namely, since in our model PVs are viewed as the stabilizers of the system, destabilization can only occur via their removal. By removing SOM inhibition onto PNs, one can increase the overall firing rate of the network and thus cortical gain, while at the same time maintaining overall network stability by freeing additional PV inhibition to PN from the weakened SOM to PV inhibitory pathway. Whereas we do not provide a plausible measure of network stability from the collected data, future studies addressing causality of our proposed roles of PVs, SOMs and VIPs on cortical rehabilitation (threshold and gain, see response to comments 2 and 3) will also address and measure the stability issue.

Comment 6: Second, the general model is based on the sound-evoked responses recorded from the soma of each neuron population, but the most pivotal parameter is synaptic strength at each connection. It is well beyond the scope of this study to collect this information, but these properties should be considered in more depth.

Response: We agree with the Reviewer that the synaptic strengths are key parameters in the computational model. However, as Reviewer #3 points out since we use the computational model to generate guidance in driving experimental approaches in understanding recovery after NIHL, we focused on exploring the vast parameter space related to this damage (strength of incoming connections) and corresponding recovery (direct recovery currents), which resulted in over 200K parameter sets (see Methods, Parameter sweep, and stability criteria). Meanwhile, our ratio of excitatory to inhibitory neurons (~20%) and the probability of connections across populations was taken to be in line with experimental results¹⁴ while the rest of the computational parameters were adjusted from previous modeling works that considered a similar circuit¹³. The Methods section failed to reference this work, so this information is now added (see Methods, Modeling noise-induced damage and recovery subsection). Nonetheless, ongoing studies in our laboratories, which are outside the scope of this manuscript, are currently addressing the synaptic mechanisms underlying cortical rehabilitation after noise trauma.

Similar to this other modeling work, we considered homogeneous synaptic strengths across the four populations, while allowing for heterogeneity to arise via differences in the probability of connections (see Methods, Mean-field and diffusion approximation subsection and Table 4). In reality, these synaptic strengths are not only heterogeneous across and within the population, but also exhibit different forms of short-term plasticity. For example, PNs to PV neurons exhibits short-term depression, while PNs to VIP neurons show short-term facilitation¹⁵.

So, while the model was instrumental in providing insight into recovery after NIHL, such shortcomings will be addressed in future work to create a more biologically realistic computational model of A1. We have added the following sentences to the discussion section to highlight this point in the manuscript.

After NIHL, whether the cortex lies in the ISN regime or remains wired to resist instability despite no longer requiring it, remain an open question. The answer likely resides in the dynamics and relative strengths of synaptic connections. As a result, such a model should account for possible heterogeneities of synaptic strengths across and within populations, as well as the diversity of

short-term plasticity rules found across different cortical synaptic connections¹⁵. Therefore, in combination with computational modeling, ongoing and future *ex vivo* and *in vivo* studies in our laboratories will address the synaptic mechanisms underlying cortical rehabilitation after noise trauma. However, it is clear that in addition to synaptic mechanisms, the cell-type-specific intrinsic plasticity found in PVs¹⁶ and VIPs (**Fig. 9**), but not in either PNs¹⁶ or SOMs (**Fig.5**), are consistent with contributions of intrinsic plasticity mechanisms in cortical recovery.

Minor comments:

Comment 7: The rationale for providing both the acute widefield and longitudinal data sets of threshold responses from A1 PNs (Figs 2d vs k) is unclear, especially since they are not consistent. For example, there is a 40 dB shift at 1 day for panel d, but only a 10 dB shift at 1 day for panel k. Another example is that there is partial recovery at 10 days for panel d, but complete recovery for panel k. How do you interpret the differences between the two data sets? I would have assumed that the longitudinal data is more robust and interpretable than the widefield data. In fact, the manuscript notes several limitations of the widefield imaging data (lines 167-170) which serve as a rationale for collecting the superior individual neuron data. It is not clear what is learned from the widefield data that is critical to the questions being addressed. What would be lost if only the L2/3 data were to be presented?

The same concern emerges for Figure 4 (PV neuron widefield vs longitudinal 2-photon imaging; panels d vs k), Figure 5 (SOM neuron widefield vs longitudinal 2-photon imaging; panels d vs k), and Figure 8 (VIP neuron widefield vs longitudinal 2-photon imaging; panels d vs k).

Response: To address this comment, we added the following to the discussion section.

Strengths and limitations of our wide-field and 2P imaging, and studies addressing different neurons and neuronal compartments

We employed A1 wide-field transcranial imaging and A1 longitudinal 2P imaging of L2/3 somata to identify the plasticity in the different neuronal cell types of A1 after NIHL. Although we found overall similar plasticity rules via 2P and wide-field imaging after NIHL, wide-field imaging showed larger shifts and reduced recovery in the response thresholds compared to the 2P imaging. The main reason that likely explains these differences is that wide-field imaging assesses responses from different neuronal compartments (somata and dendrites) and different cortical layers, whereas 2P imaging assesses only L2/3 somatic responses¹⁷. As such, wide-field imaging results might provide insights into the cell-type-specific plasticity of the entire A1 after NIHL. For example, a robust shift and lower recovery in the wide-field response thresholds after NIHL is consistent with the notion that A1 dendrites and/or neurons residing in the cortical layers other than L2/3 may undergo more robust and longer lasting changes than changes occurring in L2/3 somata after NIHL. In this context, one limitation of our study is that we studied cellular mechanisms based on distinguishing PNs, PV, SOM, and VIP neurons; however, these neuronal subtypes are not a homogeneous population^{6,18}. Moreover, our 2P studies were only limited on L2/3. Therefore, our future studies will expand to investigate different cortical neuronal subtypes in terms of their gene expression profile, projection targets, and AC layer localization.

Comment: The introduction gives the impression that recovery of synaptic function is a common occurrence following hearing loss, so it is worth clarifying the types of hearing loss that are being considered when making global statements. Both developmental hearing loss and presbycusis are each associated with many permanent changes, including to excitatory and inhibitory synapses.

Response: We revised the Introduction section to clarify that we are considering hearing loss induced by exposure to loud noises or ototoxic compounds.

Reviewer #3

We thank the Reviewer for the supportive comments and for raising important concerns. Reviewer 3 evaluated that “*This is probably the first study to carry out such extensive investigations of roles of different cell types in cortical rehabilitation, thus is a valuable addition to the scientific literature. Experiments are in general well executed. The reviewer has some specific points to improve the manuscript*”. Below we cite the raised specific points and provide our point-by-point responses.

Major:

Comment 1. The computation models the authors used are useful and generate guidance to look into specific features of cortical cell types with experimental approaches. In the four-population model, the VIP population is designed to receive feedforward input just like PN and PV cells. However, the Ji et al. (Cerebral Cortex 2016) study shows that VIP neurons (even in layer 2/3) do not receive direct thalamic input. Will removing this feedforward input affect the result?

Response: Indeed, the major strength of the modeling in this work is its use to generate testable predictions that guide the next set of experiments. For this reason, we simplified some aspects of the model (e.g., the drive onto VIPs) to minimize the number of free parameters. Nonetheless, we agree with the reviewer’s concern, especially considering the results of Ji et al. (2016).

Although VIPs lack direct thalamic input, recent papers focused on the primary mouse visual cortex suggest that VIPs receive excitatory inputs from deeper cortical layers (e.g., L4 onto L2/3) and feedback from higher-order processing areas^{19,20}. Further, there is evidence that their recurrent excitatory inputs exhibit short-term facilitation¹⁵. Together, it is reasonable to assume that these extraneous components yield excitatory inputs onto VIPs that grow in proportion to the strength of the stimulus, as we have modeled. To test this claim directly, we modified the model by removing the direct stimulus input and allowing the strength of excitatory input onto VIPs ($w_{VIP,PN}$) to vary with stimulus strength. We found that VIPs recover quantitatively very similarly, thus supporting our conclusions/results on the proposed role of VIPs. We report our results (**new Supplement Fig. 5**), along with the following paragraph to the result section.

In our four-population model (**Fig. 7a**), to simplify the parameter space, we included a direct, stimulus-dependent excitatory input onto VIPs. Since there is no evidence so far that VIPs receive direct thalamic input²¹, this input could be viewed as a simplification of feedback excitatory inputs from higher order processing, deeper cortical layers, and short-term facilitating connections from recurrently connected PNs^{15,19,22}. Removing the direct stimulus input and allowing the strength of excitatory input onto VIPs to vary with stimulus strength produces model results that are quantitatively very similar to those where VIPs receive direct stimulus inputs (**Supplement Fig. 5**, see also **Figure 9** on the role of intrinsic VIP plasticity). In total, these modeling results from the

four-population model provide a clear, testable hypothesis: VIP neurons show a strong recovery after noise trauma.

Comment 2. This manuscript implies that the recovery of VIP neuron responses is the main driving force for the increased gain of PN neurons via suppressing SOM neurons. However, it fell short of explaining what factors drive the recovery of VIP neurons. The authors can do similar ex vivo recording experiments in VIP neurons after NIHL, at least to provide some possible mechanisms.

Response: To address this concern, we performed the requested experiments (**new Fig. 9**). Namely, to explore the mechanism underlying the enhanced VIP activity after noise trauma, we compared VIP intrinsic excitability between sham- and noise-exposed mice at 10 days after trauma. Namely, after localizing the AC (**Fig. 9ab**, Methods), we performed brain slice electrophysiology experiments in AC L2/3 VIPs to assess their intrinsic excitability (**Fig. 9**, Methods). Although the resting membrane potential, AP width and threshold, and adaptation ratio did not change, we found increased input resistance (**Fig. 9d**) and firing rate (**Fig. 9j**) in VIPs from noise-exposed mice. Although we can't exclude the potential contributions of synaptic changes, this result indicates increased VIP neuron excitability 10 days after NIHL and is consistent with the notion that this increased intrinsic excitability might contribute to the enhanced recovery of VIP sound-evoked responses.

Minor:

Comment: Please provide subtitles in the Result section.

Response: Added

Comment: In Fig.2f, please label with marks to indicate whether there is a significant reduction of AN gain after NIHL.

Response: Labelled

Comment: Fig. S2j, Fig. S4g, please label the X-axis.

Response: Added

Comment: Fig.3f, what is the unit for the X-axis?

Response: Added

Comment: How old are these animals? Please indicate in the Methods.

Response: Please see the response for the major comment from Reviewer #1.

Comment: In the figure legends, brackets are not consistent, for example sometimes “a)” is used, and some others “(a)” is used. In the line 1266, where is the figure (p)? It should be figure (i).

Response: We fixed it. Thanks for the comment.

Comment: Line 47, “...due to increased cortical gain, the sensitivity of neuronal responses against sound levels.”. Add “which is” before “the sensitivity...”.

Response: added

Comment: Line 161, “...remained increased elevated...”. Remove “elevated”. Line 444, “cochlear damage may contribute to the hearing problems peripheral damage...”. Add “after” before “peripheral damage”.

Response: Revised

Reviewer #4

We thank the Reviewer for the positive comments and for providing constructive suggestions. Reviewer 1 evaluated the *“Although cortical recovery following peripheral damage has already been described, the underlying mechanisms are not known, and this study adds relevant, new, and important information to the topic. The experimental approaches are diverse and carefully conducted, and the results are solid and convincing. The study is also very well explained and follows logical steps of reasoning. The manuscript is well written. The only weak point of the manuscript is the interpretation of the results. As such, they are “just” suggestions, but a few additional experiments could significantly strengthen them.* Below we cite the suggestions and provide our point-by-point responses.

Major comments

Comment 1. The possible interpretation, as suggested for example on l.29-30 of the abstract, would be much more convincing if data could prove this. This circuit relies on the canonical circuit illustrated in Figure 7a – but the reality in an auditory cortical column is much subtler than this, and strongly dependent on context (see the review by Studer et al., Neuro Bio Rev 2021 for some examples). If manipulating VIP neurons (for example by inhibiting them with optogenetics) would lead to activity similar to pre-NE data, I would be much more convinced that this is indeed a possible mechanism. If such an experiment is not possible, I would advise to keep a very interpretative version of this mechanism (the formulation on l.299 (...support the notion that...) is preferred over the one of l.303 (...contributes to..)).

Response: Thank you very much for this important comment that had a helpful impact on the whole manuscript. Please see our response to comment 2 from reviewer 1, which addresses the general issue of interpretation of our results throughout the manuscript and our revisions in

response to this. In the revised manuscript, we clearly state the focus, the strengths and limitations, and the interpretation of our study in this context.

Comment 2. The in vitro data on SOM neurons is very nice and convincing, but I wonder why putative changes in intrinsic properties have only been explored in SOM neurons, and not in the 3 other cell types? I would have been particularly interested in knowing whether VIP neurons change their intrinsic properties upon NIHL. If they do not change, could the authors speculate about the origin of the increased activity in VIP neurons (that could then explain the changes in activity in the 3 other cell types)? This would complete the suggested mechanism, for example discussed on l.488-490.

Response: To address this concern, we performed the requested experiments (**new Fig. 9**). Namely, to explore the mechanism underlying the enhanced VIP activity after noise trauma, we compared VIP intrinsic excitability between sham- and noise-exposed mice at 10 days after trauma. Namely, after localizing the AC (**Fig. 9ab**, Methods), we performed brain slice electrophysiology experiments in AC L2/3 VIPs to assess their intrinsic excitability (**Fig. 9**, Methods). Although the resting membrane potential, AP width and threshold, and adaptation ratio did not change, we found increased input resistance (**Fig. 9d**) and firing rate (**Fig. 9j**) in VIPs from noise-exposed mice. Although we can't exclude the potential contributions of synaptic changes, this result indicates increased VIP neuron excitability 10 days after NIHL and is consistent with the notion that this increased intrinsic excitability might contribute to the enhanced recovery of VIP sound-evoked responses.

Comment 3. Given that it is widely known that the inhibitory neurons include VIP, SOM, and PV neurons (as correctly cited on l.60-62), I wonder why the model did not include VIP neurons from the beginning. In other words, what is the reasoning for having a model without VIP neurons (Figure 3), instead of starting directly with a model including this interneuron subtype (Figure 5)?

Response: We agree that we could have included the VIP neurons in the model to begin with. However, to simplify the experimental design and the manuscript narrative, we included the PV and SOM neurons first and then expand to VIP neurons.

Minor comments

Comment 1. Figures 4 and 5: Although similar, the wide-field imaging (Fig 4e and 5e) does not give the same results as the 2P responses (Fig 4m and 5m). Could the authors speculate why this would be the case?

Response: **Response:** To address this comment, we added the following to the discussion section.

Strengths and limitations of our wide-field and 2P imaging, and studies addressing different neurons and neuronal compartments

Please see the response for reviewer#1, comment 7, which includes this section.

Comment 2: L.26-27. Why not put PN, PV, and VIP together in these 2 lines?

Response: Revised

Comment 3: L.47. What is sensitivity of neuronal responses “against sound levels”?

Response: We meant neuronal response gain. We have revised the wording.

Comment 4: L.62. It would be appropriate to also cite a recent review on inhibitory neurons particularly in the auditory system: Studer & Barkat, Neuroscience and Biobehavioral Reviews 2021

Response: Cited

Comment 5: The sentence in l.91-95 would be easier to read if it would be shorter (or separated in several sentences).

Response: Revised

Comment 6: L.140 Fig. 2a, b instead of Fig. 2ab

Response: Revised

Comment 7: L.159. Although it might be in the methods, it would be nice to here briefly mention how the response gain of sound-evoked activity is quantified.

Response: Revised. We now state this in results section.

We calculated gain as the average change in the fluorescence signals ($\Delta F/F\%$) per 5 dB SPL step starting from response threshold as employed previously^{7,9,23}.

Comment 8: L. 179. Similarly, it would be nice to briefly explain how tuning strength is quantified.

Response: Revised. We now state this in results section.

To identify the sound-responsive neurons, we used a tone sensitivity index, d-prime (d'), which reflects the neurons' selectivity for preferred frequency against non-preferred frequency^{24,25} (Methods).

Comment 9. L.230. What is an “extensive, brute force parameter sweep”?

Response: To address this comment, we added clarification to the main text and the Methods sections to address this point.

Main text: Because our major focus is to understand the circuit pathways participating in the recovery of PN's threshold, high gain, and stability after NIHL, we utilized a mean-field circuit theory, which captures the average neuronal firing rate for each of the subpopulations. This allowed us to perform an extensive parameter sweep (**Fig. 3c, d**; see Methods for additional details).

Methods: We then perform an extensive sweep in the (β^{PN} , β^{PV} , $I_{\text{recov}}^{\text{PN}}$, $I_{\text{recov}}^{\text{PV}}$, $I_{\text{recov}}^{\text{SOM}}$) parameter space. More specifically, for each of these parameters, we consider n ($n = 15$ for β^{PN} and β^{PV} and $n = 10$ for the I recovery parameters) equally spaced points between bounds defined in supplement Table 3 and formed a 5-dimensional hypercube consisting of 225,000 interior points/parameter sets. We then estimate the firing rate and gain in the same way as the default case for each parameter set.

References (also included in the revised manuscript)

- 1 Erway, L. C., Shiao, Y. W., Davis, R. R. & Krieg, E. F. Genetics of age-related hearing loss in mice. III. Susceptibility of inbred and F1 hybrid strains to noise-induced hearing loss. *Hearing research* **93**, 181-187 (1996). [https://doi.org/10.1016/0378-5955\(95\)00226-x](https://doi.org/10.1016/0378-5955(95)00226-x)
- 2 Henry, K. R. & Chole, R. A. Genotypic differences in behavioral, physiological and anatomical expressions of age-related hearing loss in the laboratory mouse. *Audiology* **19**, 369-383 (1980). <https://doi.org/10.3109/00206098009070071>
- 3 Willott, J. F. Effects of aging, hearing loss, and anatomical location on thresholds of inferior colliculus neurons in C57BL/6 and CBA mice. *Journal of neurophysiology* **56**, 391-408 (1986). <https://doi.org/10.1152/jn.1986.56.2.391>
- 4 Johnson, K. R. *et al.* Effects of Cdh23 single nucleotide substitutions on age-related hearing loss in C57BL/6 and 129S1/Sv mice and comparisons with congenic strains. *Scientific reports* **7**, 44450 (2017). <https://doi.org/10.1038/srep44450>
- 5 Kane, K. L. *et al.* Genetic background effects on age-related hearing loss associated with Cdh23 variants in mice. *Hearing research* **283**, 80-88 (2012). <https://doi.org/10.1016/j.heares.2011.11.007>
- 6 Harris, K. D. & Shepherd, G. M. The neocortical circuit: themes and variations. *Nature neuroscience* **18**, 170-181 (2015). <https://doi.org/10.1038/nn.3917>
- 7 Chambers, A. R. *et al.* Central Gain Restores Auditory Processing following Near-Complete Cochlear Denervation. *Neuron* **89**, 867-879 (2016). <https://doi.org/10.1016/j.neuron.2015.12.041>
- 8 Chambers, A. R., Salazar, J. J. & Polley, D. B. Persistent Thalamic Sound Processing Despite Profound Cochlear Denervation. *Frontiers in Neural Circuits* **10** (2016). <https://doi.org/10.3389/fncir.2016.00072>
- 9 Resnik, J. & Polley, D. B. Cochlear neural degeneration disrupts hearing in background noise by increasing auditory cortex internal noise. *Neuron* (2021). <https://doi.org/https://doi.org/10.1016/j.neuron.2021.01.015>

- 10 McGill, M. *et al.* Neural signatures of auditory hypersensitivity following acoustic trauma. *eLife* **11** (2022). <https://doi.org:10.7554/eLife.80015>
- 11 Kumar, M., Xiong, S., Tzounopoulos, T. & Anderson, C. T. Fine Control of Sound Frequency Tuning and Frequency Discrimination Acuity by Synaptic Zinc Signaling in Mouse Auditory Cortex. *The Journal of neuroscience : the official journal of the Society for Neuroscience* **39**, 854-865 (2019). <https://doi.org:10.1523/jneurosci.1339-18.2018>
- 12 Ferguson, K. A. & Cardin, J. A. Mechanisms underlying gain modulation in the cortex. *Nat Rev Neurosci* **21**, 80-92 (2020). <https://doi.org:10.1038/s41583-019-0253-y>
- 13 Bos, H., Oswald, A.-M. & Doiron, B. Untangling stability and gain modulation in cortical circuits with multiple interneuron classes. *bioRxiv*, 2020.2006.2015.148114 (2020). <https://doi.org:10.1101/2020.06.15.148114>
- 14 Pfeffer, C. K., Xue, M., He, M., Huang, Z. J. & Scanziani, M. Inhibition of inhibition in visual cortex: the logic of connections between molecularly distinct interneurons. *Nat Neurosci* **16**, 1068-1076 (2013). <https://doi.org:10.1038/nn.3446>
- 15 Campagnola, L. *et al.* Local connectivity and synaptic dynamics in mouse and human neocortex. *Science (New York, N.Y.)* **375**, eabj5861 (2022). <https://doi.org:10.1126/science.abj5861>
- 16 Henton, A., Zhao, Y. & Tzounopoulos, T. A role for KCNQ channels on cell-type-specific plasticity in mouse auditory cortex after peripheral damage. *The Journal of neuroscience : the official journal of the Society for Neuroscience* (2023). <https://doi.org:10.1523/jneurosci.1070-22.2023>
- 17 Waters, J. Sources of widefield fluorescence from the brain. *eLife* **9** (2020). <https://doi.org:10.7554/eLife.59841>
- 18 Tremblay, R., Lee, S. & Rudy, B. GABAergic Interneurons in the Neocortex: From Cellular Properties to Circuits. *Neuron* **91**, 260-292 (2016). <https://doi.org:10.1016/j.neuron.2016.06.033>
- 19 Billeh, Y. N. *et al.* Systematic Integration of Structural and Functional Data into Multi-scale Models of Mouse Primary Visual Cortex. *Neuron* **106**, 388-403.e318 (2020). <https://doi.org:10.1016/j.neuron.2020.01.040>
- 20 Yao, Z. *et al.* A taxonomy of transcriptomic cell types across the isocortex and hippocampal formation. *Cell* **184**, 3222-3241 e3226 (2021). <https://doi.org:10.1016/j.cell.2021.04.021>
- 21 Ji, X. Y. *et al.* Thalamocortical Innervation Pattern in Mouse Auditory and Visual Cortex: Laminal and Cell-Type Specificity. *Cerebral cortex (New York, N.Y. : 1991)* **26**, 2612-2625 (2016). <https://doi.org:10.1093/cercor/bhv099>
- 22 Yao, S. *et al.* A whole-brain monosynaptic input connectome to neuron classes in mouse visual cortex. *Nature neuroscience* **26**, 350-364 (2023). <https://doi.org:10.1038/s41593-022-01219-x>
- 23 McGill, M. *et al.* Neural signatures of auditory hypersensitivity following acoustic trauma. *bioRxiv*, 2022.2005.2024.493204 (2022). <https://doi.org:10.1101/2022.05.24.493204>
- 24 Cody, P. A. & Tzounopoulos, T. Neuromodulatory mechanisms underlying contrast gain control in mouse auditory cortex. *The Journal of Neuroscience*, JN-RM-2054-2021 (2022). <https://doi.org:10.1523/jneurosci.2054-21.2022>
- 25 Romero, S. *et al.* Cellular and Widefield Imaging of Sound Frequency Organization in Primary and Higher Order Fields of the Mouse Auditory Cortex. *Cerebral cortex (New York, N.Y. : 1991)* **30**, 1603-1622 (2020). <https://doi.org:10.1093/cercor/bhz190>

REVIEWERS' COMMENTS

Reviewer #1 (Remarks to the Author):

The revisions are extensive and a highly responsive to the initial reviews. I would be comfortable with publishing the revised manuscript in its present form. The following comments are provided only for informational purposes.

C57Bl6J mice

I agree with the authors primary assertion which is that the study design included sham-exposed C57 mice which would control for strain. However, my point is a bit more nuanced: if both treatment groups contain a mutation that exacerbates the effects of noise exposure, then some of the results may not apply to strains that have robust ears. Specifically, the authors state that "we conducted our experiments in 8-12 weeks-old mice, and at that age group C57/B6 mice do not show an age-related hearing loss¹⁻⁵." However, there are measurements in the literature that demonstrate a functional decline by 8 weeks (see Figure 6 in <https://pubmed.ncbi.nlm.nih.gov/17559088/> and Figure 2 in <https://pubmed.ncbi.nlm.nih.gov/1759567/>). Furthermore, noise exposure during the 8-12 week interval accelerates presbycusis (Figures 1 and 2 in <https://pubmed.ncbi.nlm.nih.gov/35197842/>). I interpret these findings as evidence for a cochlea that begins to display degenerative changes by 8 weeks and that has a heightened vulnerability to noise, as compared to other mouse strains.

Behavioral results

The authors have performed a valuable new behavioral experiment (Rebuttal Figure 1), and I would support publication of these findings in the final manuscript, along with the authors conclusion "that cortical recovery might contribute to the recovery of perceptual detection thresholds after noise trauma."

New recordings

The new recordings that take into account best frequency are a terrific addition to the paper and, more broadly, to the hearing loss literature. The data set a high bar for future studies and I very much appreciate this addition to the study.

Relationship between network stability and behavior

Since the concept of network stability is pivotal to understanding the significance of the present findings, I would value a more direct discussion of this concept. The authors acknowledge that the manuscript does not present "a plausible measure of network stability" is not presented, so I would favor either removing the concept from manuscript, or providing explicit measures that could, in the future, help to explain the relationship of these findings to stability.

Reviewer #3 (Remarks to the Author):

The authors have addressed my questions. I recommend publication.

Reviewer #4 (Remarks to the Author):

The revisions answer most of my concerns, and I appreciate the new data on VIP interneurons intrinsic excitability. Although I understand the authors' arguments for removing the perceptual results from this study, I still think they were an interesting and important part of the work. Despite this, the new version of the manuscript still addresses a relevant question that has not been answered previously. The study has been thoroughly conducted, and the results are solid and convincing. It is an important addition to the literature, and I congratulate the authors for their work.

Response to Reviewers

We thank the reviewers for the positive and enthusiastic comments. We also appreciate the suggestions, which have helped us further improve the paper. Our responses are included in our point-by-point response.

Reviewer #1

We thank Reviewer 1 for the constructive and helpful comments. Reviewer 1 evaluated the “*The revisions are extensive and a highly responsive to the initial reviews. I would be comfortable with publishing the revised manuscript in its present form. The following comments are provided only for informational purposes*”. To address these comments, we revised the manuscript accordingly. Below we cite the raised comments and provide our point-by-point responses.

Minor Comments

Comment 1: C57Bl6J mice

I agree with the authors primary assertion which is that the study design included shamexposed C57 mice which would control for strain. However, my point is a bit more nuanced: if both treatment groups contain a mutation that exacerbates the effects of noise exposure, then some of the results may not apply to strains that have robust ears. Specifically, the authors state that “we conducted our experiments in 8-12 weeks-old mice, and at that age group C57/B6 mice do not show an age-related hearing loss¹⁻⁵.” However, there are measurements in the literature that demonstrate a functional decline by 8 weeks (see Figure 6 in <https://pubmed.ncbi.nlm.nih.gov/17559088/> and Figure 2 in <https://pubmed.ncbi.nlm.nih.gov/1759567/>). Furthermore, noise exposure during the 8-12 week interval accelerates presbycusis (Figures 1 and 2 in <https://pubmed.ncbi.nlm.nih.gov/35197842/>). I interpret these findings as evidence for a cochlea that begins to display degenerative changes by 8 weeks and that has a heightened vulnerability to noise, as compared to other mouse strains.

Response: To address this comment we added following section in the methods (noise exposure section).

C57/B6 background exhibits age-related late-onset hearing loss. However, we conducted our experiments in 8-12 weeks-old mice, and C57/B6 mice do not show age-related hearing loss at this age group¹⁻⁵. Importantly, we used aged-matched sham-exposed mice as controls for all time points and manipulations. Moreover, sham-exposed mice that did not show any changes in either ABR (**Fig. 1e, h-j**) or cortical response thresholds between P60 and P80 (**Fig. 2k, 4k, 5k, and 8k and supplementary fig. 2b, 3a, 4a, and 6a**), further supporting the absence of age-related hearing loss in our experiments. Taken together, although both treatment groups contain a mutation that could potentially affect vulnerability to noise exposure at some later time point, the different results we observed between sham- and noise-exposed mice, which form the basis of our conclusions, are solely due to the noise exposure, as all other factors, such as age and genetics are equal.

Comment 2: Behavioral results

The authors have performed a valuable new behavioral experiment (Rebuttal Figure 1), and I would support publication of these findings in the final manuscript, along with the authors conclusion "that cortical recovery might contribute to the recovery of perceptual detection thresholds after noise trauma."

Response: We appreciate the positive comments. However, we opted to keep the behavioral perceptual data out of this manuscript and instead include them in a future manuscript. In this future manuscript, we plan to address the causal role of the cell-type-specific cortical plasticity on the behavioral/perceptual recovery, which is not addressed at all in this manuscript. Namely, we plan to inactivate cortex, as the Rev#1 recommended in the previous review, and manipulate the activity of principal neurons and interneurons to study the role of these neuronal populations on the behavioral/perceptual recovery after noise trauma.

Comment 3: New recordings

The new recordings that take into account best frequency are a terrific addition to the paper and, more broadly, to the hearing loss literature. The data set a high bar for future studies and I very much appreciate this addition to the study.

Response: We thank the reviewer for the positive comment.

Comment 4: Relationship between network stability and behavior

Since the concept of network stability is pivotal to understanding the significance of the present findings, I would value a more direct discussion of this concept. The authors acknowledge that the manuscript does not present "a plausible measure of network stability" is not presented, so I would favor either removing the concept from manuscript, or providing explicit measures that could, in the future, help to explain the relationship of these findings to stability.

Response: To address this comment we added following text to the discussion.

Although we do not present a measure of network stability, future studies could utilize optogenetic perturbations to further investigate our proposed roles of PVs, SOMs and VIPs, while also evaluating how network stability changes after NIHL. Namely, different light intensities could be utilized to suppress a varying fraction of PVs, SOMs, and VIPs. Runaway excitation would arise while targeting the subpopulation responsible for stabilizing the network. Moreover, the light intensity at which runaway activity occurs can be used as a proxy for network stability, with higher light intensities indicating greater stability. To assess how the network stability changes during recovery, the same experiment can be repeated before and at different times after NIHL. Taken together, our observations are consistent with the notion, albeit not tested here, that PVs act as the stabilizers whereas SOMs act as the modulators of A1 plasticity.

Reviewer #3

Comment: The authors have addressed my questions. I recommend publication.

Response: We thank the reviewer for the positive comment.

Reviewer #4

Comment: The revisions answer most of my concerns, and I appreciate the new data on VIP interneurons intrinsic excitability. Although I understand the authors' arguments for removing the perceptual results from this study, I still think they were an interesting and important part of the work. Despite this, the new version of the manuscript still addresses a relevant question that has not been answered previously. The study has been thoroughly conducted, and the results are solid and convincing. It is an important addition to the literature, and I congratulate the authors for their work.

Response: We thank the reviewer for the positive comments. Regarding the comment on perceptual data, see also response to Comment 2 from Reviewer 1.

References (also included in the revised manuscript)

- 1 Erway, L. C., Shiau, Y. W., Davis, R. R. & Krieg, E. F. Genetics of age-related hearing loss in mice. III. Susceptibility of inbred and F1 hybrid strains to noise-induced hearing loss. *Hearing research* **93**, 181-187, doi:10.1016/0378-5955(95)00226-x (1996).
- 2 Henry, K. R. & Chole, R. A. Genotypic differences in behavioral, physiological and anatomical expressions of age-related hearing loss in the laboratory mouse. *Audiology* **19**, 369-383, doi:10.3109/00206098009070071 (1980).
- 3 Willott, J. F. Effects of aging, hearing loss, and anatomical location on thresholds of inferior colliculus neurons in C57BL/6 and CBA mice. *Journal of neurophysiology* **56**, 391-408, doi:10.1152/jn.1986.56.2.391 (1986).
- 4 Johnson, K. R. *et al.* Effects of Cdh23 single nucleotide substitutions on age-related hearing loss in C57BL/6 and 129S1/Sv mice and comparisons with congenic strains. *Scientific reports* **7**, 44450, doi:10.1038/srep44450 (2017).
- 5 Kane, K. L. *et al.* Genetic background effects on age-related hearing loss associated with Cdh23 variants in mice. *Hearing research* **283**, 80-88, doi:10.1016/j.heares.2011.11.007 (2012).